# The spliceosome-associated protein CWC15 promotes miRNA biogenesis in Arabidopsis

Bangjun Zhou[1,2], Huihui Yu[1,2], Yong Xue[1,2], Mu Li[1,2], Chi Zhang[1,2] & Bin Yu [1,2] ✉

MicroRNAs (miRNAs) play a key role in regulating gene expression and their biogenesis is precisely controlled through modulating the activity of microprocessor. Here, we report that CWC15, a spliceosome-associated protein, acts as a positive regulator of miRNA biogenesis. CWC15 binds the promoters of genes encoding miRNAs (MIRs), promotes their activity, and increases the occupancy of DNA-dependent RNA polymerases at MIR promoters, suggesting that CWC15 positively regulates the transcription of primary miRNA transcripts (pri-miRNAs). In addition, CWC15 interacts with Serrate (SE) and HYL1, two key components of microprocessor, and is required for efficient pri-miRNA processing and the HYL1-pri-miRNA interaction. Moreover, CWC15 interacts with the 20 S proteasome and PRP4KA, facilitating SE phosphorylation by PRP4KA, and subsequent non-functional SE degradation by the 20 S proteasome. These data reveal that CWC15 ensures optimal miRNA biogenesis by maintaining proper SE levels and by modulating pri-miRNA levels. Taken together, this study uncovers the role of a conserved splicing-related protein in miRNA biogenesis.

microRNAs (miRNAs), ~21 nucleotides in size, are important regulator of gene expression and play crucial roles in various biological processes in many eukaryotes including plants and animals[1–5]. miRNAs are generated from their primary transcripts (pri-miRNAs), which contain one or more miRNA-residing imperfect stem-loops[1–5]. Upon production, miRNAs are loaded into the effector protein called Argonaute (AGO) to direct complementary sequence-dependent post-transcriptional gene silencing[1–5].

In plants, the DNA-dependent RNA polymerase II (Pol II) transcribes most pri-miRNAs from genes encoding miRNAs (MIRs)[6]. pri-miRNAs are co-transcriptionally processed by the RNase III enzyme DICER-LIKE 1 (DCL1) in the nucleus to release a miRNA/miRNA* duplex[7,8]. The resulting miRNA duplex is subsequently methylated by Hua enhancer 1 (HEN1), and then miRNA is selectively sorted into AGO1 with the help of HEAT SHOCK PROTEIN 90 (Hsp90) and CYCLOPHILIN 40 (CyP40)[9–12]. The double stranded RNA-binding protein hyponastic leaves 1 (HYL1) and the zinc finger protein serrate (SE) form a complex with DCL1 known as the Dicing-body (D-body) to ensure efficient and accurate processing of pri-miRNAs[13–16]. The formation of D-body

depends on the disorder regions of SE, which mediate phase separation[17]. In addition, pri-miRNA processing also occurs in the nucleoplasm[18]. Interestingly, many splicing-related factors also directly or indirectly associate with the DCL1 complex to modulate miRNA biogenesis. These factors include protein phosphatase 4 (PP4) regulatory subunit 3[19], THP1 in the TREX-2 complex[20], and RBV[21], SMA1[22], cap-binding protein 80/20 (CBP80/20)[23,24], stabilized 1 (STA1)[25], AAR2[26], several proteins in the MOS4-associated complex (MAC)[27–31], serrate-associated protein 1 (SEAP1)[32] and the high expression of osmotically response gene 15/histone deacetylase 9[33]. These factors regulate pri-miRNA processing, transcription and/or stability, suggesting the connection between the splicing machinery and the DCL1 complex.

Intriguingly, both HYL1 and SE are subjected to post-translational control to fine-tune miRNA biogenesis[34,35]. The activity and stability of HYL1 is modulated by phosphorylation and dephosphorylation[26,36–42]. Dephosphorylated HYL1, but not phosphorylated HYL1, promotes miRNA biogenesis[36]. Moreover, phosphorylated HYL1 appears to be more stable in extended periods of darkness and is reactivated by

[1]Center for Plant Science Innovation, University of Nebraska-Lincoln, Lincoln, NE 68588–0666, USA. [2]School of Biological Sciences, University of Nebraska-Lincoln, Lincoln, NE 68588–0118, USA. ✉e-mail: byu3@unl.edu

dephosphorylation, suggesting that phosphorylated HYL1 may serve as a reservoir for dephosphorylated HYL1[40]. SE contains intrinsically disordered regions (IDRs) and is typically assembled into the macromolecular complexes for its functions[43]. The unloaded or excessive SE is degraded by 20 S proteasome alpha subunit G1 (PAG1) to ensure proper function of SE[43]. Interestingly, degradation of SE is triggered by the pre-mRNA processing 4 kinase A (PRP4KA)-mediated phosphorylation, which also reduces SE affinity to HYL1 and negatively affects SE-mediated miRNA production[44]. However, how SE phosphorylation and degradation are regulated is still unclear.

Cwf_Cwc_15 domain-containing protein (CWC15) from yeast and its ortholog AD002 from human were identified as a spliceosome-associated protein and later were shown as a component of the core machinery of the spliceosome[45–47]. AD002 is associated with the conserved hPrp19/CDC5L complex[45], which is required for catalytic activation of the spliceosome[48]. However, CWC15 was not found in the NineTeen Complex (NTC), the yeast counterpart of PRP19[45]. CWC15/AD002 is essential for the development of yeast and metazoans[49]. In Arabidopsis, lack of CWC15 also causes embryo lethality, while downregulation of CWC15 results in growth defects accompanied with moderate splicing defects[50]. However, the functional mechanism of CWC15 is still poorly understood.

Here, we show that CWC15 plays important roles in miRNA biogenesis. Knockdown of CWC15 with an artificial miRNA (amiR$^{CWC15}$) reduced the accumulation of miRNAs and pri-miRNAs. CWC15 binds MIR promoters, and promotes Pol II occupancy at MIR promoters, suggesting that CWC15 positively regulates pri-miRNA transcription. In addition, CWC15 interacts with HYL1 and SE, but not DCL1, facilitates HYL1-pri-miRNA interaction and is required for efficient pri-miRNA processing. Intriguingly, SE protein levels are increased in amiR$^{CWC15}$, which is caused by decreased 20 S proteasome activity. Consistent with this observation, CWC15 interacts with both 20 S proteasome and with PRP4KA, and promotes SE phosphorylation and degradation. These results suggest that CWC15 may serve as a platform to facilitate degradation of unpacked or excessive SE by promoting its phosphorylation, which ensures proper miRNA biogenesis. Taken together, this study shows that CWC15 plays multifaceted roles in miRNA biogenesis.

## Results

### CWC15 may function in miRNA biogenesis

To better understand the mechanisms governing miRNA biogenesis, we analyzed the network of CDC5, which is a core component of MAC, and promotes pri-miRNA transcription and processing[28], using the STRING program (https://string-db.org) with a high confidence score[51]. We expected to identify candidate proteins involved in miRNA biogenesis, based on the assumption that functional related proteins often co-exist in the same network. CDC5 network consists of known MAC components and several proteins involved in RNA metabolisms (Supplementary Fig. 1). Among candidate proteins, we focused on a CWC15 (EMB2769; AT3G13200), (Supplementary Fig. 1), because its potential interaction with the MAC complex[50]. CWC15 is a conserved protein in eukaryotes (Supplementary Fig. 2a) and contains a Cwf_Cwc_15 domain of unknown function (Supplementary Fig. 2b). Protein structure prediction found that the Cwf_Cwc_15 domain contains a disordered region and a coiled-coil motif (Supplementary Fig. 2b).

To evaluate its function, we first examined the association of CWC15 with MAC3A, CDC5, MAC5A using a bimolecular fluorescence complementation (BiFC) assay. In this assay, we transiently co-expressed CWC15 fused with the N-terminal fragment of yellow fluorescent protein (nYFP) with PRL1, CDC5, MAC3A or MAC5A fused with the C-terminal fragment of YFP (cYFP), in the leaves of *Nicotiana benthamiana* (*N. benthamiana*). Co-expression of nYFP-CWC15 with cYFP tagged MAC components resulted in YFP signals (Supplementary

Fig. 2c), suggesting the possible association of CWC15 with the MAC complex. Interestingly, MAC5 formed speckles with CWC15 while CDC5 and MAC3A did not. A possible explanation is that MAC5, but not MAC3A and CDC5, contains unstructured region, which is known to promote speckles formation through liquid-liquid phase separation[52]. Moreover, CWC15 also co-IPed with MAC5A (Supplementary Fig. 2d). Taken together, these results confirm that CWC15 is correlated with MAC and may function in miRNA biogenesis.

### CWC15 is required for miRNA accumulation

Next we tested if CWC15 is required for miRNA accumulation. Because lack of CWC15 causes embryo lethality[50], we used an artificial miRNA (amiR$^{CWC15}$) targeting 5′ UTR (Untranslated region) to knockdown the transcript levels of CWC15 in Col (wild-type; WT) (Fig. 1a). In T1 generation, we identified amiR$^{CWC15}$ −1, −2 and −3 lines, in which the transcript levels of CWC15 were reduced (Supplementary Fig. 3a). These lines displayed pleiotropic development defects such as smaller plant size and serrated leaves (Fig. 1b and Supplementary Fig. 3b). Expression of a CWC15 open reading frame fused with a GFP tag at its C-terminal driven by its native promoter (pCWC15::GFP-CWC15) in amiR$^{CWC15}$−1 line fully rescued the developmental defects of amiR$^{CWC15}$ (Supplementary Fig. 3b), demonstrating that the growth defects of amiR$^{CWC15}$ are caused by reduced CWC15 transcript levels.

We next examined the effect of amiR$^{CWC15}$ on miRNA accumulation by Northern blot. All examined miRNAs were reduced in abundance in two amiR$^{CWC15}$ lines relative to Col (Fig. 1c). In addition, stem-loop RT-qPCR[53] analysis showed that the reduced miRNA levels were fully recovered in the complementation lines (Fig. 1d), suggesting that CWC15 is required for miRNA accumulation. Next, we analyzed the effect of CWC15 on miRNA accumulation at global levels by comparing miRNA profile in amiR$^{CWC15}$ with that in Col through Illumina deep sequencing analyses. The result showed that many miRNAs displayed reduced accumulation in amiR$^{CWC15}$ line (Fig. 1e and Supplementary dataset 1) with a median log2 fold change −0.5 relative to Col. Taken together, these results demonstrate that CWC15 is required for miRNA accumulation.

We further performed reverse transcription quantitative PCR (RT-qPCR) to examine the transcript levels of several miRNA target transcripts including CMT3, ARF8, APS3, CKB3, PHV and SPL3/9, which are targets of miR823, miR167, miR395, miR397, miR166 and miR156, respectively. The transcript levels of these targets were moderately increased in amiR$^{CWC15}$ relative to Col and recovered in the complementation lines (Fig. 1f), agreeing with the decreased levels of miRNAs in amiR$^{CWC15}$.

### CWC15 promotes pri-miRNA transcription

Next, we asked how CWC15 promotes miRNA accumulation. We first examined the impact of CWC15 on pri-miRNA accumulation, which is one of the factors determining miRNA levels. RT-qPCR analysis showed that all examined pri-miRNAs were reduced in abundance in amiR$^{CWC15}$ relative to Col (Fig. 2a and Supplementary Fig. 4). Moreover, the pri-miRNA levels were fully recovered in the complementation lines (Supplementary Fig. 4a), suggesting that CWC15 enhances pri-miRNA accumulation. Because pri-miRNA levels are partially determined by transcription, we asked if CWC15 could affect MIR promoter activity using a transgenic line harboring a GUS gene driven by the MIR167a promoter (pMIR167a::GUS), which has been used as a reporter for MIR transcription[23]. We crossed pMIR167a::GUS into amiR$^{CWC15}$ and compared GUS expression levels in amiR$^{CWC15}$ with those in WT. GUS staining and RT-qPCR showed that GUS levels were reduced in amiR$^{CWC15}$ (Fig. 2b, c), suggesting that CWC15 may modulate MIR promoter activity. To validate the result, we next test the effect of CWC15 on the occupancy of Pol II at MIR promoters using chromatin immunoprecipitation (ChIP) assay with anti-RPB2 (the second largest subunit of Pol II) antibodies. The results showed that amiR$^{CWC15}$ reduced

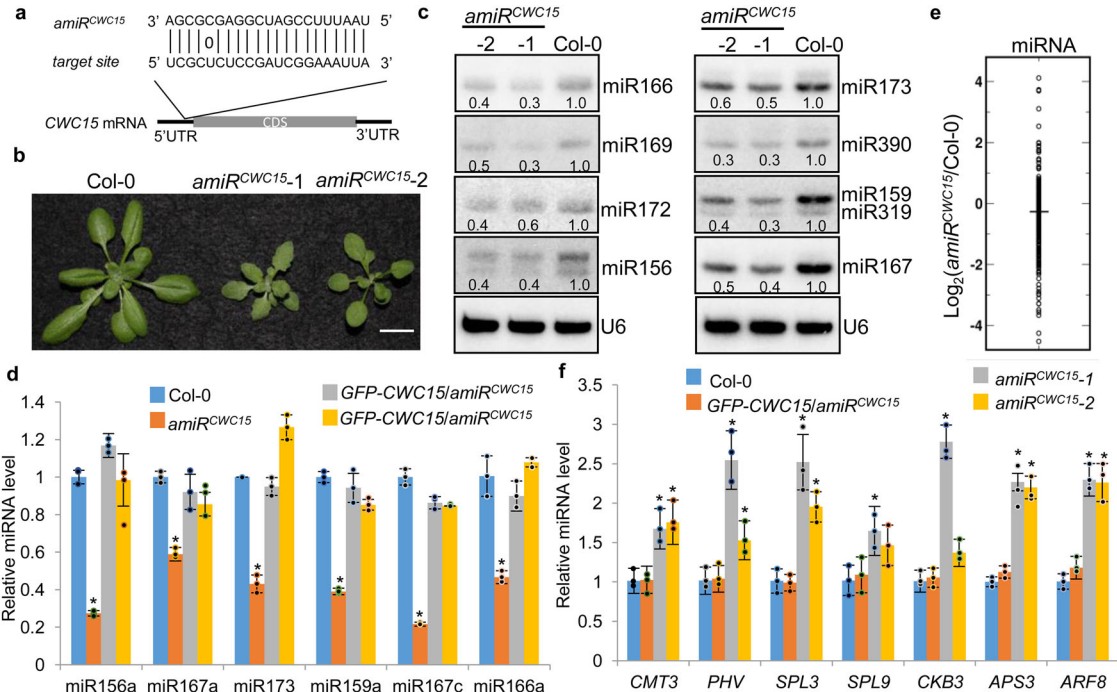

**Fig. 1 | CWC15 is required for miRNA accumulation. a** Diagram of the artificial miRNA targeting CWC15 (amiR^CWC15). Solid line represents Watson–Crick pairing and a "0" indicates a C-U mismatch pair. CDS: coding sequence; 5'UTR: 5' untranslated region; 3'UTR: 3' untranslated region. **b** Seedlings of Col and amiR^CWC15 (Scale bar: 1 cm). **c** miRNA levels in various genotypes detected by Northern blot. U6 RNA serves as the loading control. The numbers shown below the pictures indicate the relative amounts of miRNAs (the values in Col were set as 1). **d** miRNA levels in various genotypes detected by RT-qPCR. miRNA levels were normalized to U6. Error bars indicate SD from three biological replicates (*n* = 3) and data are presented as mean values +/- SD. Asterisks indicate significantly reduced expression of amiRCWC15 compared to Col based on *P*-values (two-tailed unpaired *t*-test) (*P* < 0.01): miR156a, 5.52677E-06; miR167a, 0.000834; miR173, 3.13908E-05;

miR159a, 6.66758E-06; miR167c, 6.62854E-06; miR166a, 0.001156458. **e** Small RNA sequencing analysis in Col and amiR^CWC15–1. The miRNA abundance was calculated as reads per million, and a Log2-transformed ratio of amiR^CWC15/Col-0 was plotted. Each circle represents one miRNA. *N* = 375 distinct miRNAs. **f** The transcript levels of miRNA target genes in various genotypes detected by RT-qPCR. The levels of miRNA target transcripts were normalized to those of ACTIN2 and compared with Col. Error bars indicate SD from three biological replicates (n = 3) and data are presented as mean values +/- SD. Asterisks indicate significantly increased expression of amiRCWC15 −1/−2 compared to Col based on *P*-values (two-tailed unpaired *t*-test) (*P* < 0.05): CMT3, 0.0193/0.0164; PHV, 0.00293/0.0423; SPL3, 0.00243/ 0.00273; SPL9, −.00409/0.0738; CKB3, 0.00027/0.0532; APS3, 0.002/0.000183; ARF8, 0.000591/0.0011.Source data are provided as a Source Data file.

the occupancy of Pol II at MIR promoters (Fig. 2d). To test if CWC15 directly affects MIR transcription, we examined the interaction of CWC15 with RBP2 using co-IP. However, RBP2 did not co-IPed with CWC15 (Fig. 2e). We next examined the occupancy of CWC15 at MIR promoters by ChIP analysis of the transgenic plants harboring a p35S::2HA-CWC15 transgene with antibodies recognizing HA epitope tag. qPCR detected the enrichment of examined MIR promoter fragments in HA-CWC15 IPs (Fig. 2f). Taken together, these results show that CWC15 plays a role in regulating MIR transcription.

Because CWC15 is a splicing factor, we also examined its effect of CWC15 on pri-miRNA splicing. RT-PCR analysis showed that the splicing of some pri-miRNAs was impaired in amiR^CWC15 (Supplementary Fig. 4b).

### CWC15 interacts with the DCL1 complex
To further explore the role of CWC15 in miRNA biogenesis, we examined its association with the DCL1 complex using BiFC. YFP signals were observed in the leaves of *N. benthamiana* co-expressing nYFP-CWC15/cYFP-HYL1, nYFP-CWC15/cYFP-SE or nYFP-CWC15/cYFP-DDL but not in leaves harboring nYFP-CWC15/cYFP-DCL1 (Fig. 3a), suggesting that CWC15 may interact with, DDL, HYL1 and SE. To validate this result, we transiently co-expressed MYC-CWC15/HYL1 −2HA or MYC-CWC15/SE-2HA in *N. benthamiana* and performed IP with antibodies recognizing the MYC epitope tag. HYL1-2HA or SE-2HA, but not the negative control GFP-2HA, was enriched in the MYC-CWC15 precipitates (Fig. 3b, c). To further confirm the result, we IPed GFP-CWC15 from the complementation line harboring the pCWC15:GFP-CWC15

transgene and found that CWC15 co-IPed with HYL1 and SE (Fig. 3d). These results reveal that CWC15 interacts with HYL1 and SE.

### CWC15 promotes the cleavage activity of the DCL1 complex
The interaction of CWC15 with HYL1 and SE suggested that CWC15 might have a role in modulating DCL1 activity. We thus examined if CWC15 could modulate pri-miRNA cleavage by the DCL1 complex using an in vitro assay[54]. As previously described, [^32P] labeled pri-miR162b (MIR162b) harboring the stem-loop of miR162b with 6-nt arms at each end and pre-miR162b (Fig. 4a, b) were incubated with protein extracts from young flowers of amiR^CWC15 or Col to evaluate the production of miR162b (supplementary Fig. 5). The result showed that the amount of miR162b processed from radioactive labeled MIR162b or pre-miR162b was lower in amiR^CWC15 than in Col (Fig. 4c, d and supplementary Fig. 5). At 120 min time point, miR162b levels produced from MIR162b and pre-miR162b in amiR^CWC15 were ~20% and ~45% of those generated in Col, respectively (Fig. 4e, f), suggesting that CWC15 may promote the optimal activity of the DCL1 complex.

### CWC15 promotes the interaction of HYL1 with pri-miRNAs
The less impact of CWC15 on pre-miR162b than on MIR162b suggests that CWC15 may have additional roles in pri-miRNA processing. One possibility is that CWC15 might help the interaction of pri-miRNAs with the DCL1 complex given its interaction with SE and HYL1. We compared the association of pri-miRNAs with HYL1 in amiR^CWC15 with that in Col, which has been considered as indicator for the loading of pri-miRNA to the DCL1 complex, by RNA immunoprecipitation (RIP) using

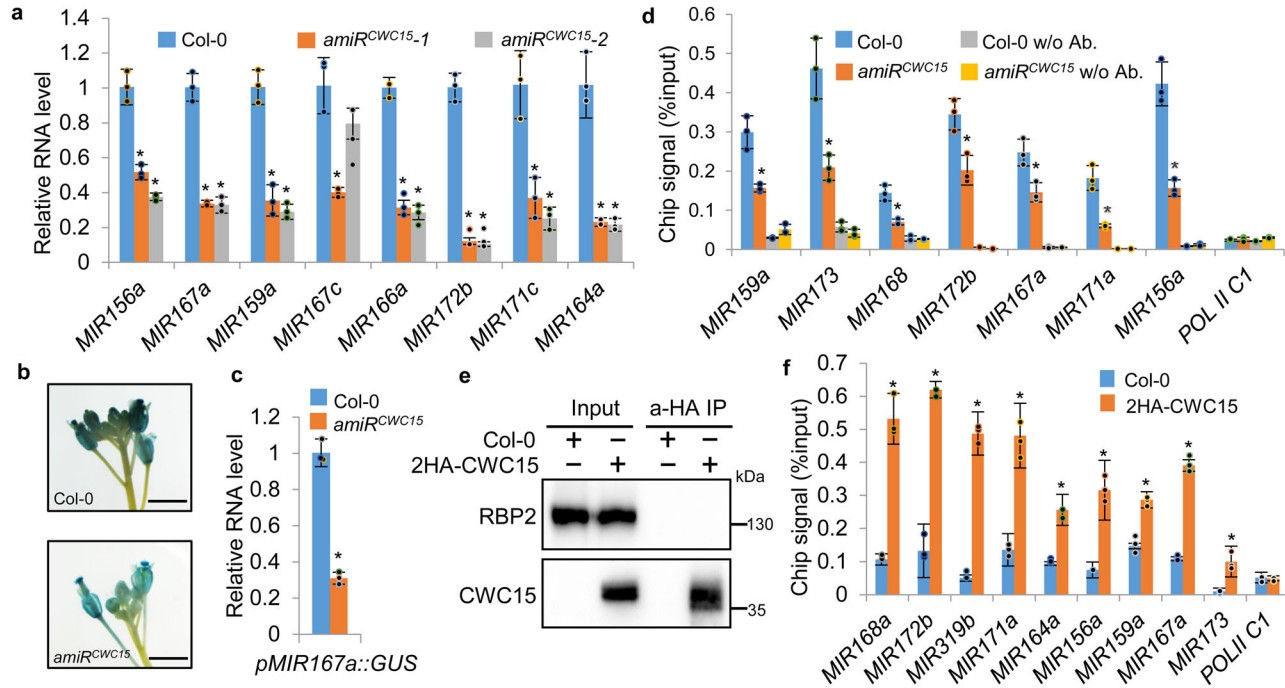

**Fig. 2 | amiRCWC15 reduces pri-miRNA accumulation through affecting transcription. a** RT-qPCR analysis of pri-miRNA levels in Col-0 and amiR[CWC15]. pri-miRNA levels were normalized to those of ACTIN2 and compared with those in Col (set as 1). Error bars indicate SD from three biological replicates ($n = 3$) and data are presented as mean values +/- SD. Asterisks indicate significantly reduced expression compared to Col based on P-value (two-tailed unpaired t-test; $P < 0.01$):. MIR156a, 0.00161/0.000479; MIR167a, 0.00146/0.00022; MIR159a, 0.001097/0.000328; MIR167c, 0.002947/0.088748;MIR166a, 0.000175/0.00011;MIR172b, 0.000113/0.000107; MIR171c, 0.007848/0.002982;MIR164a, 0.002065/0.00199.
**b** Histochemical staining of GUS in inflorescence of Col and amiR[CWC15] harboring the pMIR167a::GUS transgene. Fifteen plants for each genotype were analyzed and a picture is shown (Scale bar: 0.5 cm). **c** GUS transcript levels in the indicated plants (the values in Col were set as 1) detected by RT-qPCR. Error bars indicate SD from three biological replicates and data are presented as mean values +/- SD. w/o Ab. indicates without antibodies. Asterisks indicate significantly reduced expression compared to Col based on P-value (two-tailed unpaired t-test; $P < 0.01$): 0.000135.
**d** Pol II occupancy at MIR promoters in the indicated plants detected by ChIP

followed by qPCR. The intergenic region between At2g17470 and At2G17460 (POL II C1) was used as a negative control. Error bars indicate SD from three biological replicates ($n = 3$) and data are presented as mean values +/- SD. Asterisks indicate significantly reduced occupancy based on P-value (two-tailed unpaired t-test; $P < 0.05$): MIR159a, 0.011092; MIR173, 0.043162; MIR168, 0.024155; MIR172b, 0.000269; MIR171a, 0.023305; MIR156a, 0.000283; MIR167a, 0.000271; POL II C1, 0.788731. **e** Co-IP between CWC15 and RPB2 in Col harboring the p35S::2HA-CWC15 transgene. Co-IP was performed using anti-HA. 2HA-CWC15 and RBP2 were detected by western blot. **f** The occupancy of CWC15 at MIR promoters in the indicated plants detected by ChIP followed by qPCR. The intergenic region between At2g17470 and At2G17460 (POL II C1) was used as a negative control. Error bars indicate SD from three biological replicates ($n = 3$) and data are presented as mean values +/- SD. Asterisks indicate significant occupancy based on P-value (two-tailed unpaired t-test; $P < 0.05$): MIR168a, 0.003198; MIR172b, 0.000133; MIR319b, 0.00024; MIR171a, 0.000321; MIR164a, 0.019332; MIR156a, 0.007154; MIR159a, 0.000671; MIR167a, 0.000435; MIR173, 0.001713, Source data are provided as a Source Data file.

---

antibodies recognizing HYL1. Similar amount of HYL1 was IPed from amiR[CWC15] and Col (Fig. 4g). RT-qPCR revealed that the amounts of examined pri-miRNAs (MIR172b, MIR167a, MIR159a, MIR173, MIR156b and MIR156a) in HYL1 IPs from amiR[CWC15] were lower than those from Col (Fig. 4h).

## CWC15 promotes the degradation of SE proteins

To understand how CWC15 affects DCL1 activity, we examined the effect of CWC15 on the transcript levels of DCL1, DDL, SE, HYL1, CBP20/80 and HEN1, which function in miRNA biogenesis, through RT-qPCR. The transcript levels of these genes slightly increased in amiR[CWC15] (Supplementary Fig. 6). We further evaluated the proteins levels of DCL1, SE and HYL1 and found that SE and DCL1 protein levels were increased in amiR[CWC15] relative to those in Col, while HYL1 protein levels were not altered (Fig. 5a). Because SE is also subjected to degradation, we examined if the increased SE protein levels in amiR[CWC15] were caused by increased stability[43]. Indeed, after treatment with MG132, which is a proteasome inhibitor known to block SE degradation, the abundance of SE in Col was comparable to that in amiR[CWC15] (Fig. 5b), suggesting that the decay of SE in amiR[CWC15] may be disrupted. To validate this result, we treated cell lysates from Col and amiR[CWC15] seedlings with cycloheximide (CHX) to block protein synthesis and examined the half-life of SE. Agreeing with previous

report, the half-life of SE in Col was ~10 min (Fig. 5c). In contrast, SE half-life in amiR[CWC15] was extended to ~20 min (Fig. 5c), suggesting that CWC15 promotes the degradation of SE.

## CWC15 is required for the 20 S proteasome-mediated SE degradation

We next asked if the increased SE stability is caused by impaired 20 S proteasome activity, which degrades SE[43]. We used an in vitro 20 S proteasome assay, in which the 20 S proteasome was isolated through immunoprecipitation of the PAG1 complex[43], to test this possibility. Col plants harboring a PAG1-FIAG-4MYC transgene driven by its native promoter (pPAG1::PAG1-FM) was crossed with amiR[CWC15] to obtain amiR[CWC15] containing pPAG1::PAG1-FM. The protein levels of PAG1 in amiR[CWC15] were like those in Col (Supplementary Fig. 7a, b). We IPed the PAG1 complex (Fig. 5d). Silver stain assay showed that 20 S proteasomes were purified successfully (Fig. 5e). Moreover, the purified 20 S proteasomes was able to degrade recombinant SE protein in vitro and its activity was inhibited by MG132 (Supplementary Fig. 7c, d), agreeing with previous report[43]. Thus, we used this system to test the effect of amiR[CWC15] on the 20 S proteasome-mediated SE degradation. The result showed that the degradation rate of SE by the 20 S proteasome from amiR[CWC15] was lower than that from Col (Fig. 5d–f), demonstrating that CWC15 is required for the 20 S proteasome-mediated SE degradation.

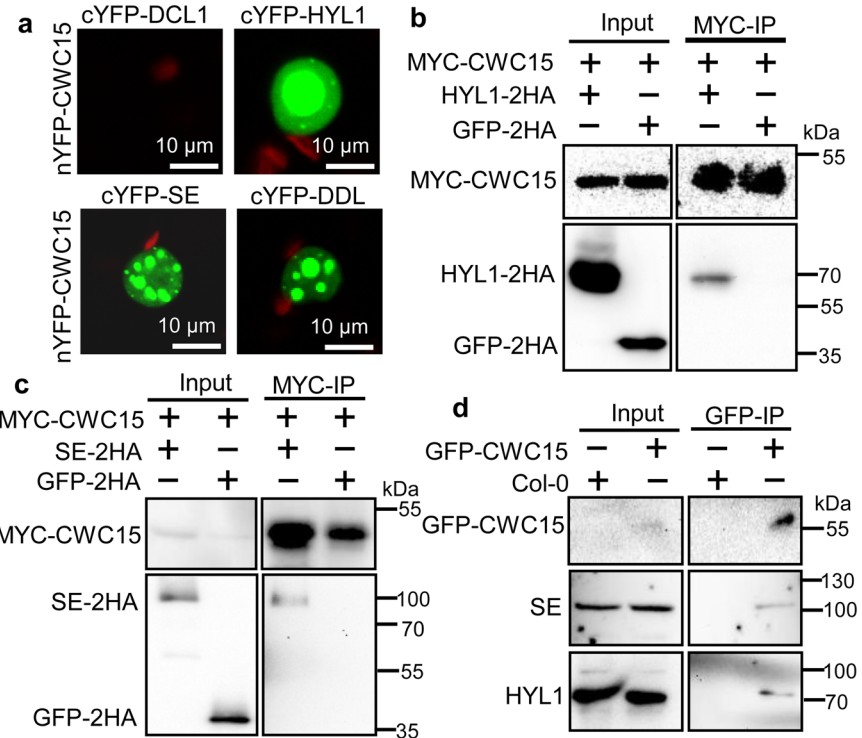

**Fig. 3 | CWC15 associates with the DCL1 complex. a** The interactions with CWC15 with DCL1, HYL1, SE and DDL analyzed by BiFC. Paired cYFP- and nYFP-fusion proteins were co-expressed in leaves of *N. benthamiana*. Green color indicates the BiFC signal detected by a confocal microscopy at 48 h after infiltration. Scale bar = 10 μm. Three independent experiments yielded consistent results. **b** Co-IP between MYC-CWC15 and HYL1−2HA. HYL1-2HA and GFP-2HA were co-expressed with MYC-CWC15 in leaves of *N. benthamiana*, respectively. IPs were performed using anti-MYC antibodies. HYL1-2HA, GFP-2HA and MYC-CWC15 were detected by western blot. Two independent experiments yielded consistent results. **c** Co-IP between MYC-CWC15 and SE-2H. SE-2HA and GFP-2HA were co-expressed with MYC-CWC15 in leaves of *N. benthamiana*, respectively. IPs were performed using anti-MYC antibodies. SE-2HA, GFP-2HA and MYC-CWC15 were detected by western blot. Two independent experiments yielded consistent results. **d** Co-IP between CWC15 and SE or HYL1 in pCWC15::GFP-CWC15/ amiR^CWC15 transgenic plants. Co-IP was performed using anti-GFP antibodies. GFP-CWC15, SE and HYL1 were detected by western blot. Two independent experiments yielded consistent results. Source data are provided as a Source Data file.

Because impaired 20 S proteasome activity leads to the accumulation of SE in cytoplasm[43], we asked if amiR^CWC15 had a similar effect using a nuclear–cytoplasmic fractionation assay. The result showed that the ratio of cytoplasm-localized SE in amiR^CWC15 was increased relative to Col, in which SE was mostly localized in the nucleus (Fig. 5g, h)

## CWC15 associates with the 20 S proteasome to promote SE degradation

The effect of CWC15 on SE degradation promoted us to test if CWC15 is associated with the 20 S proteasome using the BiFC assay. Among four selected 20 S proteasome subunits known to interact with SE[43], CWC15 interacted with PAB1, PAG1 and PBA1, but not PBE1, in the nucleus (Fig. 6a). It should be noted that CWC15-PAG1 BiFC signal could also be observed in the cytoplasm. Agreeing with this result cell fractionation assay showed that CWC15 localized at both cytoplasm and nucleus (Supplementary Fig. 8).

To validate the result, we transiently co-expressed MYC-CWC15 with PAB1-2HA, PAG1-2HA or GFP-2HA and performed co-IP with anti-MYC antibodies (Fig. 6b). MYC-CWC15 co-IPed with PAB1-2HA, PAG1-2-HA, but not GFP-2HA. Moreover, 2HA-CWC15 co-IPed with PAG1-FM in the stable transgenic line harboring PAG1-FM and 2HA-CWC15 (Fig. 6c). Taken together, these results demonstrated the association of CWC15 with 20 S proteasome.

The association of CWC15 with SE and the 20 S proteasome raised at least two possible ways by which CWC15 affects SE degradation. It may facilitate the recruitment of SE to the 20 S proteasome and/or promotes the 20 S proteasome activity on SE. In first scenario, we

would expect a reduction of SE amount in the 20 S proteasome of amiR^CWC15 relative to Col, while in the second scenario, SE will be accumulated in the 20 S proteasome of amiR^CWC15. To test these two possibilities, we IPed PAG1-FM from transgenic lines expressing PAG1-FM (Fig. 6d). Similar amount of PAG1 was IPed from Col and amiR^CWC15 (Fig. 6d). The amount of SE in the PAG1 complex from amiR^CWC15 were ~2.1 times that from Col (Fig. 6d). Moreover, the amount difference of SE in the PAG1 complex between Col and amiR^CWC15 was reduced after treatment with MG132 (Fig. 6d). These data suggest that CWC15 may promote 20 S proteasome activity on SE through its association with the 20 S proteasome and SE.

## CWC15 associates with PRP4KA to promote SE phosphorylation

Because SE degradation is triggered by PRP4KA-mediated phosphorylation, we sought to test if CWC15 promotes SE degradation by modulating phosphorylation. To examine this possibility, we first tested the interaction of CWC15 with PRP4KA using BIFC. Co-expression of cYFP-CWC15/ nYFP-PRP4KA or nYFP-CWC15/ cYFP-PRP4KA resulted in YFP signal (Fig. 7a). Moreover, CWC15 and PRP4KA reciprocally pulled down each other when co-expressed (Fig. 7b, c), showing the CWC15-PRP4KA interaction. This result promoted us to test if CWC15 affected SE phosphorylation. SE phosphorylation status was analyzed using phos-tag gels, which contain a phos-tag that binds phosphate-containing molecules resulting in reduced mobility of phosphorylated proteins. The results showed that the phosphorylated form of SE was reduced in amiR^CWC15 relative to Col plants (Fig. 7d, e). Taken together, these results show that CWC15 interacts with PRP4KA and promotes its activity on SE.

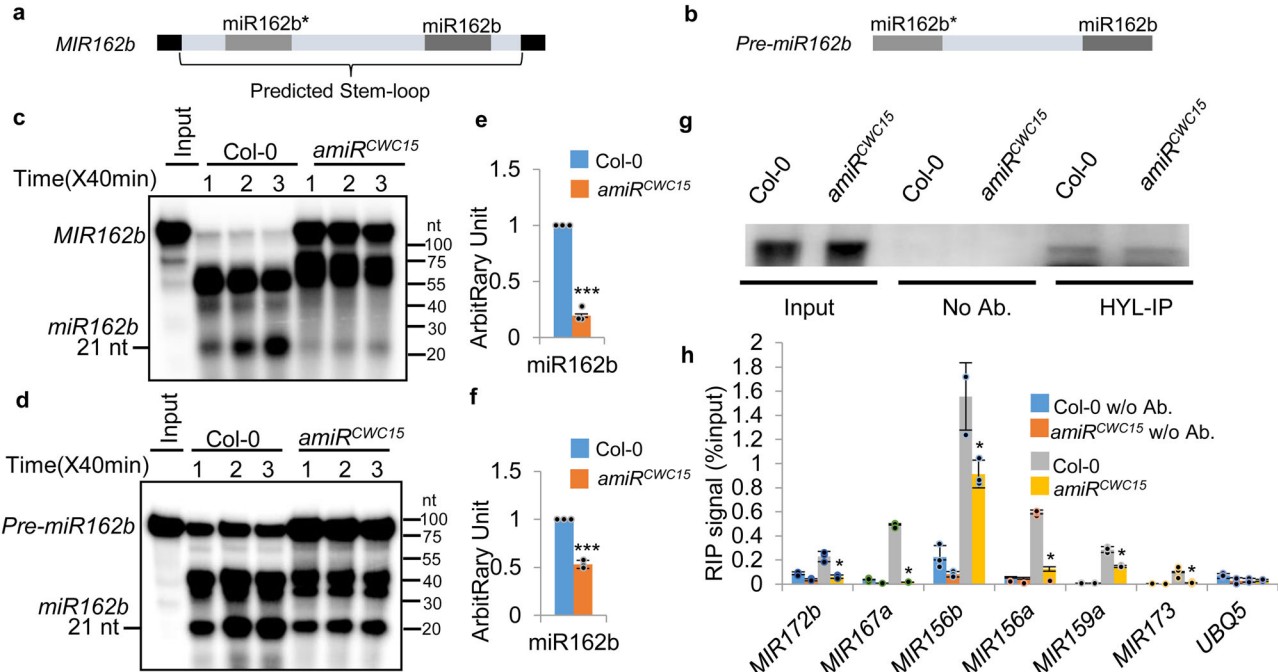

**Fig. 4 | CWC15 is required for miRNA maturation. a, b** Schematic diagram of the MIR162b (**a**) and pre-miR162b (**b**) used in vitro processing assay. **c** and **d** miR162b produced from MIR162b and pre-miR162b in proteins extracts from inflorescences of amiR^CWC15 or Col. The reactions were stopped at various time points as indicated in the picture. **e** and **f** Quantification of miR162b production in amiR^CWC15 compared to that in Col. Quantification analysis was performed at 80 min. The radioactive signal of miR162b were normalized to input and compared with that of Col-0. The amount of miR162b produced in Col was set as 1. Error bars indicate SD from two biological replicates (*n* = 2) and data are presented as mean values +/- SD. ***P < 0.001 (P = 2.66793E-05 in a; P = 4.18939E-05 in b; two-tailed unpaired *t*-test).

**g** and **h** The association of HYL1 with pri-miRNAs in amiR^CWC15 relative to Col detected by RNA immunoprecipitation. IP was performed with anti-HYL1 antibodies. HYL1 was detected by Western blot. Pri-miRNAs associated with HYL1 were examined by RT-qPCR (**h**) and normalized to the input. UBQ5 serves as a negative control. w/o Ab. indicates without antibodies. Error bars indicate SD from two biological replicates (*n* = 2) and data are presented as mean values +/− SD. Asterisks indicate significantly reduced association with pri-miRNAs compared to Col based on *P*-values (two-tailed unpaired *t*-test; *P* < 0.05): MIR172b, 0.002867; MIR167a, 0.000157; MIR156b, 0.020845; MIR156a, 0.027099; MIR159a, 0.000751; MIR173, 0.030127; UBQ5, 0.898117. Source data are provided as a Source Data file.

## Discussion

CWC15 is a conserved protein in eukaryotes spliceosome[45–47]. In yeast, CWC15 interacts with several U5 snRNA components and was suggested to stabilize the spliceosomal core[45–47]. Lack of CWC15 causes lethality in yeast, Cattle, and Arabidopsis, suggesting its essential role in development. However, the exact function of CWC15 in development and RNA metabolism including splicing remains to be identified. In this study, we find that knockdown of CWC15 results in reduced miRNA accumulation. In addition, CWC15 interacts with SE and HYL1, and binds MIR loci. These results support that CWC15 directly acts in miRNA biogenesis, which shall partially contribute to its essential role in development.

SE is an IDP containing IDRs, which act as scaffolds for formation of D-bodies[17], and needs to be assembled into complexes for its normal function[43]. Homeostasis of SE levels is crucial for its proper function[43]. In Arabidopsis, it has been shown that excess amount of or unpacked SE disrupts the formation of the D-bodies, impairing miRNA biogenesis[43]. We find that the rate of SE degradation is slower in amiR^CWC15 relative to Col. Moreover, the accumulation of SE is increased in the cytoplasmic fraction in amiR^CWC15, which resembles the observation in pag1[43], suggesting that CWC15 is required for the degradation of nonfunctional SE (Fig. 7f). In Arabidopsis, the 20 S proteasome is responsible for the degradation of SE. The reduced degradation rate of SE in amiR^CWC15 promoted us to examine the interaction between CWC15 and the 20 S proteasome. Indeed, CWC15 is associated with the 20 S proteasome. The interaction of CWC15 with both 20 S proteasome and SE raises the possibility that CWC15 may deliver SE to 20 S proteasome. However, this is unlikely the case because the accumulation of SE in the inactivated 20 S proteasome from amiR^CWC15 is not reduced compared with Col. Thus, CWC15 may

promote the activity of 20 S proteasome on SE (Fig. 7f). Efficient SE degradation requires phosphorylation by PRP4KA. We find that the ratio of nonphosphorylated to phosphorylated SE is significantly lower in amiR^CWC15. In addition, CWC15 interacts with PRP4KA. These results suggest that CWC15 facilitates SE degradation by promoting its phosphorylation (Fig. 7f). In addition, CWC15 may promote SE degradation by its interaction with the 20 S proteasome.

Previous study showed that the 20 S proteasome and PRP4KA interact with SE in the nucleus, suggesting that SE is phosphorylated by PRP4KA in the nucleus and subsequently degraded by 20 S proteosome at the nucleus[43,44]. However, SE is accumulated in cytoplasm of pag-1 and prp4ka. It was proposed that the accumulation of SE maybe due to the reprogramming of protein trafficking genes in pag-1[43]. BiFC analysis suggested that CWC15 mainly interacts with the 20 S proteasome and PRP4KA in the nucleus, showing that it may promote their activity in the nucleus. CWC15 also localizes in the cytoplasm, indicating that it may also affect 20 S proteasome-mediated reprograming of protein trafficking genes. Alternatively, it may help SE degradation in cytoplasm. It shall be noted that SE transcript levels are increased in amiR^CWC15, which may also contribute to increased SE protein levels.

SE acts as a platform to form the D-bodies and/or to recruit the D-body to pri-miRNAs, or vice versa[17,55]. We suspect that excess amount of SE in amiR^CWC15 may affect the formation of active DCL1 complex and/or the recruitment of pri-miRNA to the DCL1 complex. Indeed, pri-miRNA processing efficiency and HYL1-pri-miRNA interaction are reduced in amiR^CWC15. CWC15 may also have additional roles in modulating pri-miRNA processing. In yeast, CWC15 stabilizes the interaction between U6 RNA and PRP8[56]. By analogy, CWC15 may be able to stabilize HYL1-pri-miRNA interaction. In addition, CWC15 contains a

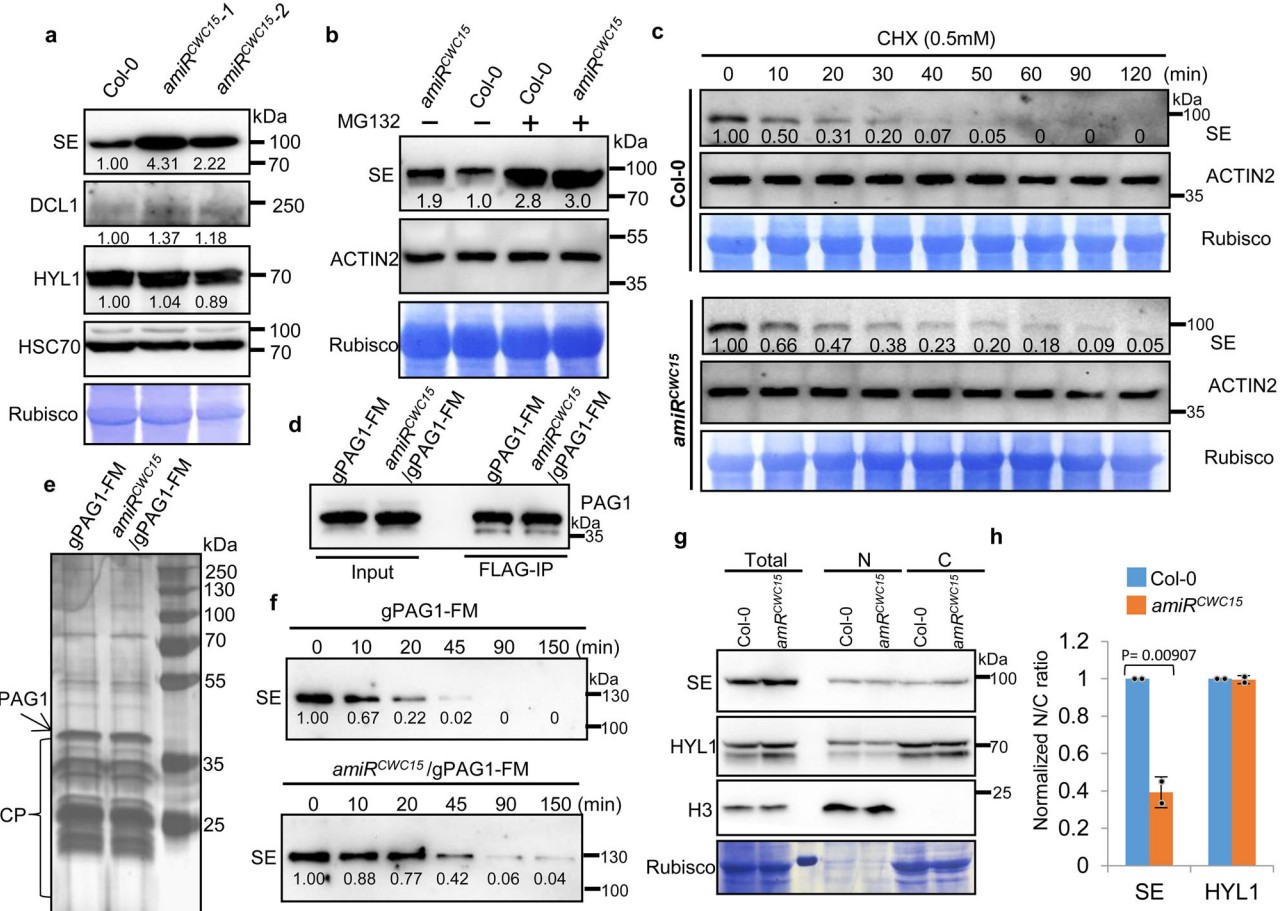

**Fig. 5 | amiRCWC15 affects the degradation of SE. a** The protein levels of DCL1, HYL1, SE detected by western blot in Col and amiR[CWC15]. Rubisco stained with Coomassie brilliant blue and HSC70 were used as loading control.DCL1 was detected by anti-DCL1(1: 1000 dilutions, rabbit polyclonal, AS194307, Agrisera). Three independent experiments yielded consistent results. **b** SE levels in Col and amiR[CWC15] seedlings with or without MG132 treatment determined by western blot. Rubisco serve as loading control. Two independent experiments yielded consistent results. **c** In vitro cell-free SE-decay assay. Total proteins from Col and amiR[CWC15] were extracted and incubated with CHX (0.5 mM) and ATP (5 mM) for the indicated times. SE levels were determined by western blot. Rubisco and ACTIN2 serve as loading control. Two independent experiments yielded consistent results. **d** PAG1 IPed from 10 day-old plants detected by western blot using an anti-FLAG antibody. IPed PAG1-FM was eluted with the FLAG peptide. Two independent experiments yielded consistent results. **e** Silver-staining of IPed PAG1-FM complex resolved in 10% SDS-PAGE. The arrow and bracket indicate PAG1-FM proteins and subunits of the 20 S core proteasome (CP), respectively. Two independent experiments yielded consistent results. **f** In vitro reconstitution assays of SE degradation via affinity

purified 20 S proteasome. Recombinant SE proteins were incubated with IPed PAG1-FM complex from Col or amiR[CWC15] harboring pPAG1::gPAG1-FM. The reaction mixture was stopped at the indicated time intervals. The numbers below the gels indicate the relative mean signals of SE proteins at different time points that were normalized to those of the proteins at time 0, where the value was arbitrarily assigned a value of 1. **g** Cell-fractionation analysis of SE protein from Col and amiR[CWC15] plants. Total extraction (T), nuclear fraction (N) and cytoplasmic fraction (C). Western blot analysis was conducted with an anti-SE or anti-HYL1 antibodies. Rubisco stained with Coomassie brilliant blue and histone 3 detected by anti-H3 antibody were used as controls for the cytoplasmic- and nuclear-specific fractions, respectively. **h** Quantification of the nuclear–cytoplasmic distribution of SE protein. The nuclear and cytoplasmic fraction ratios (N/C) of SE protein in amiR[CWC15] were sequentially normalized to that of Col-0, which was arbitrarily assigned a value of 1. Bar graph representing the relative N/C ratios derived from band intensity. Error bars indicate SD between two biological replicates ($n = 2$) with two-tailed unpaired $t$-test, $p$-values indicated above plot. Source data are provided as a Source Data file.

coiled-coil domain, suggesting that it may bind pri-miRNAs affecting miRNA biogenesis.

Pri-miRNAs are co-transcriptionally processed. The processing and transcription are coordinated by several protein factors including NOT2 (Negative on TATA less2)[57], Elongator[58], PRL1/2[27,31], CDC5[29], MAC3[29,31], MAC7[31] and SNC1 (SUPPRESSOR OF npr1-1, CONSTITUTIVE 1)[59], which interact with pol II and the DCL1 complex. CWC15 binds MIR promoter, promotes MIR promoter activity, is required for Pol II occupancy at MIR loci and interacts with the DCL1 complex strongly support that CWC15 positively regulate MIR transcription and links pri-miRNA transcription with processing (Fig. 7f). We envision that CWC15 may facilitate pri-miRNA transcription at MIR locus, where it removes unpacked/unfolded SE to help HYL1-pri-miRNA interaction and/or to promote efficient pri-miRNA processing (Fig. 7f). CWC15 does not interact with RBP2, suggesting that CWC15 may indirectly recruit Pol II. One possibility is that

CWC15 may modulate pri-miRNA transcription through its interaction with CDC5 (Fig. 7f), which directly interacts with Pol II and recruits Pol II to MIR locus[28].

We noticed that some miRNAs are upregulated in amiR[CWC15]. A possible explanation is that these miRNAs may be derived from pri-miRNAs harboring specific features. However, sequence analysis did not identify such feature. Alternatively, it could be due to the function of CWC15 in splicing, which may control the biological processes that indirectly affect the expression of specific miRNAs. Supporting this notion, the accumulation of some miRNAs also is increased in other mac mutants[27–29,31]. Interestingly, CWC5 differentially affects the accumulation of miRNAs produced from two arms of some pri-miRNAs (Supplementary dataset 1). For instance, the abundance of miR-171c-3p, miR-167a-5p and miR-167c-5p was decreased in amiR[CWC15], while the amount of miR-171-5p miR-167a-3p and miR-167c-3p were increased in

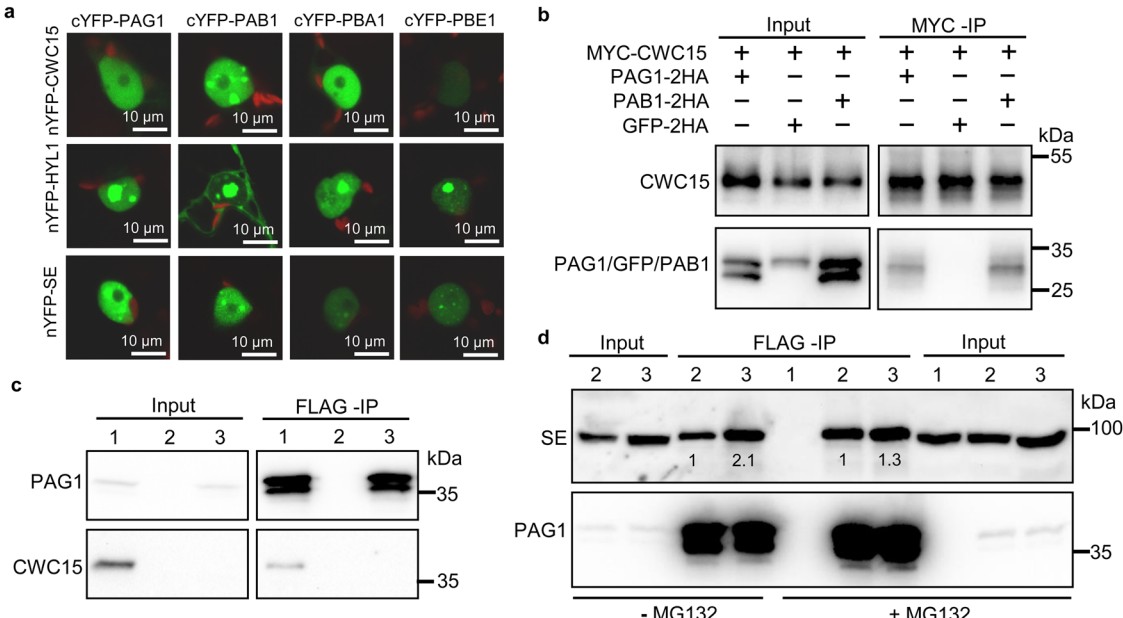

**Fig. 6 | CWC15 promotes 20 S proteasome activity. a** The protein interactions analyzed by BiFC. Paired cYFP- and nYFP-fusion proteins were co-expressed in tobacco leaves. Green color indicates the BiFC signal detected by a confocal microscopy at 48 h after infiltration. Scale bar = 10 μm. Three independent experiments yielded consistent results. **b** The association of CWC15 with PAG1 and PAB1 detected by Co-IP. PAG1−2HA, PAB1-2HA and GFP-2HA were co-expressed with MYC-CWC15 in leaves of *N. benthamiana*, respectively. IPs were performed using anti-MYC antibodies. PAG1-2HA, PAB1-2HA, GFP-2HA and MYC-CWC15 were detected by western blot. Two independent experiments yielded consistent results. **c** Co-IP between CWC15 and PAG1 in plants containing the p35S::2HA-CWC15 and

pPAG1::gPAG1-FM transgenes. 2HA-CWC15 and PAG1-FM were detected by western blot. Numbers on top of the pictures indicate plants used: 1, 2HA-CWC15/gPAG1-FM; 2, Col-0; 3, gPAG1-FM. Two independent experiments yielded consistent results. **d** The enrichment of SE in the PAG1 complex with or without MG132 treatment. IP was performed using anti-FLAG antibodies. SE and PAG1-FM were detected by western blot using anti-SE and anti-FLAG, respectively. Numbers on top of the pictures indicate plants used: 1, Col-0; 2, gPAG1-FM, 3, amiR^CWC15/gPAG1-FM. The enriched SE protein compared with IPed PAG1-FM in amiR^CWC15 was normalized to that of Col-0, which was assigned a value of 1. Two independent experiments yielded consistent results. Source data are provided as a Source Data file.

amiR^CWC15. Studies have revealed that the differential accumulation of two arms from the same precursor is controlled at processing or post-processing levels[60,61]. Given the association of CWC15 with the DCL1 complex, we suspect that CWC15 could play a role in modulating the arm-specific processing of some pri-miRNAs.

　　In summary, we uncover an activity of CWC15 outside splicing. It associates with pri-miRNA processing complex and enhances pri-miRNA processing by facilitating the degradation of excess amount of SE and promoting the interaction of HYL1 with pri-miRNAs. In addition, CWC15 positively contributes to pri-miRNA transcription by promoting the occupancy of Pol II at MIR promoters. CWC15 also functions in splicing in Arabidopsis. It may act like its counterparts from human and yeast to stabilize the splicing machinery[45–47]. In Arabidopsis, some pri-miRNAs also contain introns, whose splicing may affect processing by DCL1. It is possible that CWC15, like several other splicing factors, may also affect miRNA biogenesis through modulating splicing of some pri-miRNAs. However, the accumulation of miRNAs originated from both intron and intronless pri-miRNAs is reduced in amiR^CWC15, suggesting that the role of CWC15 in promoting miRNA biogenesis is independent of its function in splicing. CWC15 is a conserved protein in eukaryotes. Like in Arabidopsis, CWC15 might also influence the production of miRNAs in other organisms by modulating orthologs of SE, including the mammalian ARS2. Supporting this notion, the null mutations of CWC15 cause embryo lethality of mammals, which agrees with essential of miRNA in controlling development.

## Methods
### Plant materials and growth condition
Transgenic line containing pPAG1::gPAG1-FM (gPAG1-FM) is in the Columbia (Col-0) genetic background[43]. The transgenic line harboring the pMIR167a::GUS transgene[23] was crossed to amiR^CWC15. In the F2 generation, ten Col-0 and amiR^CWC15 plants containing the GUS

transgene were selected by PCR-based genotyping and pooled for GUS transcript level analyses. All plants were grown at 22 °C with 16-h light and 8-h dark cycles.

### Plasmid construction
The artificial miRNA targeting CWC15 (amiR^CWC15) was designed using WMD3[62]. A synthetic fragment containing amiR^CWC15 and MIR159a backbone was cloned into pBTEX1300. CWC15 CDS sequence directed by its native promoter was amplified by RT-PCR, cloned into pBA-eGFP to construct pCWC15::eGFP-CWC15 for complementation assay. To construct nYFP-CWC15, nYFP-SE, nYFP-HYL1 and nYFP-PRP4KA for BiFC assay, CWC15, SE, HYL1 and PRP4KA were cloned into pSPYNE(R) 173 at Kpn I site. PRL1, CDC5, MAC3A, MAC5A, AGO1, DCL1, HYL1, SE, DDL, PAG1, PAB1, PBA1, PBE1 and PRP4KA were cloned into pSPYCE (MR) at Kpn I site. For Co-IP assay, CWC15 and PRP4KA were cloned into PBA-35S-FM between Xho I and Sma I sites to construct p35S::FM-CWC15 and p35S::FM-PRP4KA. CWC15 was cloned into pBTEX1300-2HA at Sma I site to construct pBTEX1300-2HA-CWC15. HYL1, SE, PAB1 and PAG1 were cloned into pBTEX1300-2HA at Kpn I site to construct p35S::HYL1-2HA, p35S:: SE-2HA, p35S::PAB1-2HA and p35S::PAG1-2HA. The primers are listed in Supplementary Table 1.

### Phylogenetic analysis
For sequence alignment, sequences of interest in the FASTA format were input into the ClustalX 1.8 program and aligned using the ClustalX algorithm[63]. The aligned sequences were then used for phylogenetic analysis using the MEGA6 program[64]. To build an unrooted phylogenetic tree using MEGA6, the evolutionary history was inferred using the neighbor-joining method. The bootstrap consensus tree inferred from 2000 replicates was taken to represent the evolutionary history of the taxa analyzed. Branches corresponding to partitions reproduced in <50% bootstrap replicates

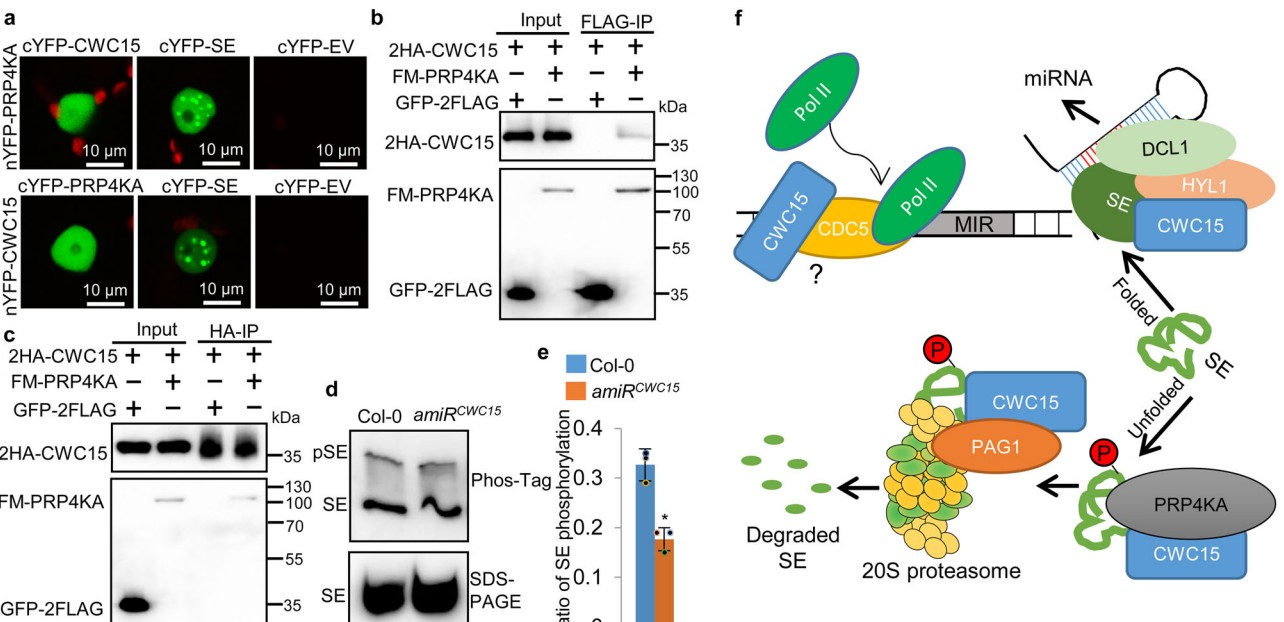

**Fig. 7 | CWC15 promotes phosphorylation of SE. a** Interaction between CWC15 and PRP4KA detected by BiFC. Paired cYFP- and nYFP-fusion proteins were co-expressed in leaves of *N. benthamiana*. Green color indicates the BiFC signal detected by a confocal microscopy at 48 h after infiltration. Scale bar = 10 μm. Three independent experiments yielded consistent results. **b**, **c** Co-IP between CWC15 and PRP4KA. 2HA-CWC15 was co-expressed with FM-PRP4KA or GFP−2FLAG in *N. benthamiana* leaves, respectively. IPs were performed using anti-FLAG antibodies or and anti-HA antibodies. 2HA-CWC15, FM-PRP4KA and GFP-2FLAG were detected by western blot. Two independent experiments yielded consistent results. **d** Phosphorylation status of endogenous SE in amiR^CWC15 and Col-0. SE was detected by SDS-PAGE and phosphor-tag gel in parallel using anti-SE antibodies.

**e** Phosphorylation levels of SE in amiR^CWC15 relative to Col. Phosphorylated SE proteins were compared with total SE proteins (set as 1) in Col and amiR^CWC15, respectively. Error bars indicate SD from two biological replicates ($n = 2$) and data are presented as mean values +/- SD. Asterisks indicate significantly reduced phosphorylation compared to Col-0 plants based on *P*-value (two-tailed unpaired *t*-test; $P < 0.01$): 0.002787. **f** The proposed action model of CWC15 in miRNA biogenesis. CWC15 binds the MIR promoters to facilitate the recruitment of Pol II, which may depend on the function of CDC5. CWC15 interacts with SE and HYL1 to enhance pri-miRNA processing. Moreover, CWC15 interacts with PRP4KA and the 20 S proteosome to promote phosphorylation-dependent SE degradation. Source data are provided as a Source Data file.

were collapsed. The evolutionary distances were computed using the Jones-Thornton-Taylor matrix-based method[65], with units representing the number of amino acid substitutions per site. The rate variation among sites was modeled with a gamma distribution (shape parameter = 1).

## Plant complementation

pCWC15::eGFP-CWC15 were transformed into amiR^CWC15 using the *Agrobacterium*-mediated floral dip method. The transgenic plants harboring pCWC15::eGFP-CWC15 were identified through screening for Basta resistance.

## Co-IP assay

The Co-IP assay was performed as previously described[66]. To examine the interactions of CWC15 with HYL1, SE, PAB1 and PAG1, MYC-CWC15 was transiently co-expressed with HYL1-2HA, SE-2HA, PAB1-2HA and PAG1-2HA in *N. benthamiana* leaves, respectively. The interaction between CWC15 and PRP4KA was performed by co-expressing 2HA-CWC15 and FLAG-PRP4KA in *N. benthamiana*. IP was performed on protein extracts using anti-HA (26181, Thermo Scientific), anti-MYC (GTA020, Bulldog Bio) or anti-FLAG (A4596, Sigma) coupled to protein G agarose beads. After IP, proteins were detected with Western blot using antibodies against MYC (1:5000 dilutions, rabbit monoclonal, M4439, Sigma), FLAG (1:10000 dilutions, mouse monoclonal, A8592, Sigma) or HA (1:5000 dilutions, mouse monoclonal, H6533, Sigma). For the interaction of CWC15 and SE or HYL1 in Arabidopsis, IP was performed with ChromoTek GFP-Trap® Agarose in pCWC15::eGFP-CWC15 transgenic plant. After IP, proteins were detected with Western blot using anti-GFP (1:1000 dilutions, rabbit polyclonal, ab190584, Abcam), anti-HYL1 (1:5000 dilutions, rabbit polyclonal, AS06136,

Agrisera) or anti-SE (1:5000 dilutions, rabbit polyclonal, AS09532A, Agrisera) antibodies.

## BiFC assay

Paired cYFP- and nYFP-fusion proteins were transiently co-expressed in in *N. benthamiana* leaves. After 48 h, YFP and chlorophyll auto-fluorescence signals were observed by a confocal microscopy (Nikon A1 HD25).

## GUS histochemical staining

For GUS staining, tissues from amiR^CWC15or WT plants harboring pMIR167a::GUS were incubated for 5 h in the staining solution at 37 °C. Tissue clearing was performed with 70% ethanol.

## Dicer activity assay

In vitro dicer activity assay was performed according to[54]. MIR162b or pre-miR162b RNA was prepared by in vitro transcription under the presence of [α-$^{32}$P] UTP. For annealing, α-$^{32}$P labeled MIR162b or pre-miR162b was heated at 95°C for 5 min in the 37.5 μl annealing buffer (20 mM Tris-HCL pH 7.5; 0.5 M NaCl; 5 mM EDTA) and then kept at ambient temperature for 1 h. Dicer protein extracts were prepared by incubating of 0.1 g flower tissue powder with 400 μl dicer extraction buffer (20 mM Tris-HCL pH 7.5; 4 mM MgCl$_2$; 5 mM DTT) at 4°C for 1 h. After spining at 13000 x g, 4°C for 15 min, the supernatants were collected and used for Dicer activity assay. 14 μl protein extracts (2 mg/ml) were mixed with 2 μl annealed MIR162b or pre-miR162b RNA and 4 μl Dicer reaction buffer (0.5 M NaCl; 5 mM ATP; 1 mM GTP; 6 mM MgCl$_2$; 125 mM creatine phosphate, 150 μg/ml creatine kinase and 6 μl RNAsin Rnase inhibitor), and reacted at ambient temperature for various time. The reaction was stopped by adding 50 μl chloroform for RNA

extraction. RNA was resolved on a 16% denaturing PAGE gels. Radioactive signals were quantified with ImageQuant (5.2).

## Northern Blot and RT-qPCR Analyses

~15 μg total RNAs extracted from inflorescences were separated on 16% PAGE (polyacrylamide gel electrophoresis) gel and transferred to nylon membranes. [32]P-labeled antisense DNA oligonucleotides were used to detect small RNAs. Radioactive signals were detected with a PhosphorImager and quantified with ImageQuant.

The transcript levels of pri-miRNAs, miRNA targets, genes involved in miRNA biogenesis and GUS mRNAs were determined using RT-qPCR. Reverse transcription was performed on DNA-free total RNAs using M-MLV reverse transcriptase (Promega) with oligo-d(T) primers according to manufacturer's instructions. The resulting cDNAs were then used as templates for qPCR on an iCycler apparatus (Bio-Rad) with the SYBR green kit (Bio-Rad). The primers used for PCR are listed in supplementary Table 1.

## RIP analyses

RIP was performed as described[67]. Three biological replicates were performed. A total of ~2 g seedlings of Col or amiR[CWC15] were cross-linked with 1% formaldehyde. After quenching with glycine, nuclei were extracted and lysed in lysis buffer (50 mM Tris-HCl, pH 8.0; 10 mM EDTA; 1% SDS; 0.1% PMSF and 1% protease inhibitor cocktail) by sonication five times with Branson Sonifier 250 of 10 s pulses of 20% power and 50 s pause between pulses. Then, equal amounts of proteins from various samples were diluted and incubated with anti-HYL1 antibodies conjugated to protein G agarose beads or protein A/G agarose beads (for no-Ab controls). After reverse cross-linking in the presence of 20 μg proteinase K (Invitrogen) at 65 °C for 1 h, RNAs were extracted and used as templates for qRT-PCR analyses, using primers listed in Supplementary Table 1.

## ChIP assay

ChIP was performed using 14-d-old seedlings from Col-0, amiR[CWC15] or amiR[CWC15] harboring p35S::2HA-CWC15 as described[27]. Three biological replicates were performed. 2 g seedlings in a 50 ml Falcon tube, were crossed linked with 37 ml of 1% formaldehyde in cross-linking solution. The reaction was stopped by adding glycine to a final concentration of 0.125 M and application of vacuum for additional 5 min. The crossed linked seedlings were ground into fine powders and suspended in 30 ml extraction buffer 1 (10 mM sodium phosphate buffer, pH7.0; 0.1 M NaCl; 10 mM β-mercapto-ethanol; M 2-methyl 2,4-pentanediol; 0.1% PMSF and 1% protease inhibitor cocktail). After filtering with two layers of Miracloth, the extracts were centrifuged for 20 min at 2100xg at 4 °C. The resulting pellets were resuspended in 1 ml of extraction buffer 2 (10 mM sodium phosphate, pH7.0; 0.1 M NaCl; 10 mM β-mercapto-ethanol; 1 M hexylene glycol; 10 mM MgCl₂; 0.5% Triton X-100; 0.1% PMSF and 1% protease inhibitor cocktail) and centrifuged at 12 000 x g for 10 min at 4 °C. The pellets were then suspended in in 360 μl of nuclei lysis buffer (50 mM Tris-HCl, pH 8.0; 10 mM EDTA; 1% SDS; 0.1% PMSF and 1% protease inhibitor cocktail), mixtured with 820 μl ChIP dilution buffer (1.1% Triton X-100; 1.2 mM EDTA; 16.7 mM Tris-HCl, pH 8.0; 167 mM NaCl; 0.1% PMSF and 1% protease inhibitor cocktail), and sonicated with Branson Sonifier 250 for 7 cycles of 10 s pulses of 20% power with 50 s pause between pulses. After centrifuging at 16,000 rpm for 10 min at 4 °C, the supernatants were collected and incubated with Anti-RPB2 (1: 5000 dilutions; mouse monoclonal, ab817, Abcam) antibodies or anti-HA antibodies (1:5000 dilutions, mouse monoclonal, H6533, Sigma) for IP. After IP, the participates were washed sequentially with low salt washing buffer (150 mM NaCl; 0.1% SDS; 1% Triton X-100; 2 mM EDTA; 20 mM Tris–HCl, pH 8.0), high salt washing buffer (500 mM NaCl; 0.1% SDS; 1% Triton X-100; 2 mM EDTA; 20 mM Tris–HCl, pH 8.0), and LiCl washing buffer (0.25 M LiCl; 1% NP-40; 1% sodium deoxycholate; 1 mM EDTA; 10 mM Tris–HCl, pH

8.0), TE buffer (10 mM Tris–HCl, pH 8.0; 1 mM EDTA). The participates were then incubated with 250ul of elution buffer (1% SDS; 0.1 M NaHCO3), at 65 °C for 15 min, followed by adding 20 ul 5 M NaCl reverse the cross-links by an overnight incubation at 65 °C. qPCR was performed on DNAs copurified with RPB2 or 2HA-CWC15, using primers listed in Supplementary Table 1.

## sRNA-seq

Small RNA libraries were prepared using total RNAs extracted from inflorescences of Col-0 or amiR[CWC15]. Two biological replicates were performed. After sequencing, miRNA analysis was performed according to[54]. For miRNA-Seq data analysis, short reads were mapped to Araport11 genome using bowtie (v0.12.7). Only uniquely mapped perfect-match reads were kept. HTSeq (v0.6.1p1) was used for read-counting on protein-coding genes (based on Araport11) and mature miRNA (from miRbase) with the strand-specific mode in HTSeq (v0.6.1p1). The total numbers of perfectly aligned reads were used for normalization[54]. The gene/miRNA read-count data were normalized with DESeq2(v1.30.1). Differential analysis of miRNA expression between two groups was conducted by DESeq2 (v1.30.1).

## In vitro 20 S proteasome activity assay

The 20 S proteasome was purified as previously described[43]. Briefly, IPs were performed on protein extracts from 14-day-old seedlings harboring gPAG1-FM with the anti-FLAG M2 magnetic bead (Sigma M8823). After washing, proteins were eluted with 250 μl of extraction buffer containing 500 ng μl⁻¹ of the 3XFLAG peptide (DYKDDDDK) by 30 min rotation at 4 °C. The 20 S proteasome-decay assays were performed as described[43]. In vitro 20 S proteasome-decay was performed on ice in 40 μl reaction mixture (150 nM purified SE protein; 10 nM Purified 20 S proteasome; 50 mM Tris-HCl (pH 7.5); 2% DMSO or 50 μM MG132) at 22 °C. The reaction was stopped by adding 2 × SDS–PAGE loading buffer at various time points (0, 10, 20, 45, 90 and 150 min).

## In vitro cell-free protein decay assay

The in vitro cell-free decay assay was carried out as described[43]. Ten-day-old seedlings of Col-0 or amiR[CWC15] were ground to a fine powder in liquid nitrogen, mixed with two fold volume of lysis buffer (25 mM Tris-HCl, pH 7.5, 10 mM NaCl, 10 mM MgCl₂ and 10% glycerol) and incubated at 4 °C for 30 min. The total protein extracts of each sample were centrifuged twice at 4 °C for 10 min at 12,000 x g and then were adjusted to equal concentrations with the lysis buffer. The final supernatant was supplemented with 0.5 mM CHX and 5 mM ATP, and the mixtures were then divided into two parts. One aliquot was added with 50 μM MG132 and the other with 2% DMSO as a control. The mixtures were then incubated at 22 °C for various times before western blot analysis.

## Nuclear−cytoplasmic fractionation assay

Nuclear−cytoplasmic fractionation was performed as described[43]. Ten-day-old seedlings of Col-0 and amiR[CWC15] were ground to a fine powder in liquid nitrogen and mixed with two volumes of lysis buffer. After the lysates were filtered through two layers of Miracloth, the flowthroughs were centrifuged at 1,500 x g for 10 min at 4 °C. The supernatant parts were then centrifuged again at 10,000 x g for 10 min at 4 °C and collected as cytoplasmic fraction. The pellet parts were washed four times with nuclear resuspension buffer 1 (NRB1, 20 mM Tris-HCl, pH 7.4, 25% glycerol, 2.5 mM MgCl₂ and 0.2% Triton X-100). After washing, the pellet was resuspended with 500 ml of NRB 2 (20 mM Tris-HCl, pH 7.5, 0.25 M sucrose, 10 mM MgCl₂, 0.5% Triton X-100, 5 mM β-mercaptoethanol and 1X protease inhibitor, Roche). Then, 500 ml of NRB 3 (20 mM Tris-HCl, pH 7.5, 1.7 M sucrose, 10 mM MgCl₂, 0.5% Triton X-100, 5 mM β-mercaptoethanol and 1Xprotease inhibitor) was carefully added on the top of samples, followed by centrifuging at 16,000 x g for 45 min at 4 °C. The final pellet was resuspended in 400 ml of lysis buffer and collected

as nuclear fraction. The quality of fractionation was validated with cytoplasmic (Rubisco) and nuclear (H3) markers.

## Phos-tag analysis
SE phospho-isoforms were analyzed using phos-tag SDS-PAGE following the manufacturer's instructions (Wako). Briefly, total proteins were separated on a phos-tag SDS-PAGE gel and then transferred to a PVDF membrane. SE was then detected with the SE antibodies.

## Reporting summary
Further information on research design is available in the Nature Portfolio Reporting Summary linked to this article.

## Data availability
The data supporting the findings of this study are available from the corresponding authors upon request. All Sequencing data used in this study are available in National Center for Biotechnology Information Gene Expression Omnibus under accession codes: GSM7122973 GSM7122974 GSM7122975 and GSM7122976. Araport11 genome (https://www.arabidopsis.org/download/index-auto.jsp?dir=%2Fdownload_files%2FGenes%2FAraport11_genome_release) was used for miRNA-Seq data analysis. Source data are provided with this paper.

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

## Acknowledgements

This work is supported by grants from National Institute of Health (GM127414 to B.Y.) and National Science Foundation (National Science Foundation (MCB-1818082 to B.Y. and C.Z.)). We thank Xiuren Zhang for sharing pET28a-Avi-6×His-SUMO-SE vector and pBA002a-P^PAG1^-gPAG1-FM seeds.

## Author contributions

B.Y. and B.Z. conceived this research, designed experiments and analyzed the data. B.Z. performed the majority of the experiments. H.Y. and C.Z. performed the sRNA-seq data analysis. Y.X. and M.L. directed the microscopy analysis. B.Y. and B.Z. drafted the manuscript. All authors approved the final version of the manuscript.

## Competing interests

The authors declare no competing interests.
