## [Peer Review File · Nature Communications]

The spliceosome-associated protein CWC15 promotes miRNA biogenesis in ArabidopsisREVIEWER COMMENTS

Reviewer #1 (Remarks to the Author):

The authors have conducted a highly intriguing study on CWC15, a component of the splicing complex, that plays roles in miRNA biogenesis. The findings indicate that CWC15 acts as a positive regulator, and its deficiency leads to reduced accumulation of pri-miRNAs and miRNAs. Moreover, CWC15 directly interacts with major microprocessor components, SE and HYL1, and it plays a role in controlling the half-life of SE by promoting its phosphorylation and degradation. Overall, the study sheds light on a new layer of the miRNA biogenetic pathway, providing valuable insights into the regulatory mechanisms involving CWC15, SE, and the 20S proteasome. However, to strengthen the conclusions and further contribute to the research field, some minor revisionary experiments are suggested. Addressing the functional implications of these findings will be crucial for obtaining a comprehensive understanding of the cellular processes governed by CWC15 and its impact on SE homeostasis and miRNA biogenesis. The research has the potential to advance the field of miRNA biogenesis and RNA processing. Consequently, addressing these points below will be of value and interest to the plant science community.

1. In Supplementary Figure 2, the BiFC assay results indicate that CWC15 appears to be associated with CDC5 and MAC3A, with the formation of speckles observed with MAC5A. However, the authors only confirmed the results of the CWC15/MAC5A complex using a co-immunoprecipitation (co-IP) assay. The authors need to provide an additional explanation for the interactions between CWC15 and CDC5 or MAC3A and how they differ from the interaction with MAC5A.
2. In Figure 1d, the expression levels of miRNAs were determined with a certain qRT-PCR method, however, it does not seem to monitor the miRNA expression but rather seems to monitor precursors because the method description did not reflex the proper method used. To ensure that the miRNA expression levels presented in Figure 1d are indeed reflective of mature miRNAs, the authors should clearly state the specific miRNA qRT-PCR method used and provided. This information is crucial for a reliable and accurate assessment of miRNA expression in the study.
3. In Figure 1e, it would be beneficial for the authors to include the median value of miRNA expression to facilitate the discussion of the overall reduction ratio and whether CWC15 functions as a general positive factor or a selectively functional factor.
4. In supplementary Figure 3, the expression levels of GFP-CWC15 should be presented to confirm that the complementation line.
5. In Figure 2c-2f, the CWC15-dependent modulation of the miR167a promoter is monitored using GUS analysis and qRT-PCR. However, to confirm the specific interaction between CWC15 and the miR167a promoter, it is essential to include ChIP (Chromatin Immunoprecipitation) data.
6. In the processing assay, the size of the cleaved intermediates between MIR162b and pre-miR162b appears to be different. To confirm whether these are the same fragments, it would be beneficial to include RNA size markers in the analysis.
7. The authors' claim that CWC15 is involved in SE degradation (Figure 5) may face challenges based on the observations in Figure S5. The transcript level of SE was notably increased by over two-fold, and the Chase assay indicated that deficiency of CWC15 extended the half-life of SE by only 10 minutes. However, a reconstituted 20S proteasome degradation assay demonstrated a more substantial extension of SE half-life to approximately 35 to 40 minutes (Figure 5f). This suggests that the up-regulation of SE transcripts could be a contributing factor to the result observed in Figure 5c. To clarify this point, the authors should provide further explanation and delve into the relationship between the increased transcript levels and the extended SE half-life, addressing any potential mechanisms that might be responsible for the observed effects.
8. In Supplementary Figure S6d, there appears to be an error in the image-Q. The sample treated for 20 minutes shows a much lower SE (signal intensity) level compared to the sample treated for 30 minutes. However, the quantitation values for both samples are reported as the same, which seems inconsistent.
9. In Figure 5, the authors stated that HYL1 is not altered in amiRCWC15 (Figure 5a). However, upon

careful observation, it appears that the HYL1 protein level seems to exhibit a slight accumulation in Figure 5g (Total). Additionally, DCL1 also appears to show a slight increase in the artificial mutants in Figure 5a. These findings might raise questions and warrant further explanation from the authors. To address this, the authors should provide a more detailed explanation of these specific findings or present clearer and more conclusive results that align with their descriptions in the figure.

10. CWC15 is a splicing-related gene. Does CWC15 influence the splicing of miRNA or other genes, like SEAP1(SERRATE-associated protein)?

11. In supplementary Figure 5, it would be beneficial to include the P-value in the legend. This addition will provide essential statistical information and enhance the clarity of the results presented in the figure.

12. Indeed, the localization of CWC15 in the nucleus is consistent with its role as a splicing factor. The BiFC assay also supports this nuclear localization, suggesting its involvement in nuclear processes. However, the detection of an association between CWC15 and the 20S proteasome in the nucleus raises questions regarding the regulation of SE degradation by CWC15 in the cytoplasm. Considering that impaired 20S proteasome activity leads to the accumulation of SE in the cytoplasm, it becomes less clear how CWC15, which is primarily localized in the nucleus, might be involved in regulating SE degradation in the cytoplasm. This observation raises the need for further investigations to better understand the interplay between CWC15, SE, and the 20S proteasome in different cellular compartments. Additional experiments, such as Co-IP assay in specific subcellular fractions, could help to shed light on the mechanisms involved in the regulation of SE degradation and how CWC15 might play a role in this process.

13. The recent study reporting the dual role of the HOS15/HDA9 complex in both promoter activity and pri-miRNA processing under ABA may add relevance to the introduction and could potentially support the role of CWC15 and its additional role in SE proteostasis.

Point-by-Point Response to Reviewer's comments

We really appreciate the insightful suggestions from both referees, which greatly improve the quality of this research. The changes in the text are highlighted in Blue.

Reviewer #1 (Remarks to the Author):

The authors have conducted a highly intriguing study on CWC15, a component of the splicing complex that plays roles in miRNA biogenesis. The findings indicate that CWC15 acts as a positive regulator, and its deficiency leads to reduced accumulation of pri-miRNAs and miRNAs. Moreover, CWC15 directly interacts with major microprocessor components, SE and HYL1, and it plays a role in controlling the half-life of SE by promoting its phosphorylation and degradation. Overall, the study sheds light on a new layer of the miRNA biogenetic pathway, providing valuable insights into the regulatory mechanisms involving CWC15, SE, and the 20S proteasome. However, to strengthen the conclusions and further contribute to the research field, some minor revisionary experiments are suggested. Addressing the functional implications of these findings will be crucial for obtaining a comprehensive understanding of the cellular processes governed by CWC15 and its impact on SE homeostasis and miRNA biogenesis. The research has the potential to advance the field of miRNA biogenesis and RNA processing. Consequently, addressing these points below will be of value and interest to the plant science community.

Really appreciate the positive comments and wonderful suggestions!

1, In Supplementary Figure 2, the BiFC assay results indicate that CWC15 appears to be associated with CDC5 and MAC3A, with the formation of speckles observed with MAC5A. However, the authors only confirmed the results of the CWC15/MAC5A complex using a co-immunoprecipitation (co-IP) assay. The authors need to provide an additional explanation for the interactions between CWC15 and CDC5 or MAC3A and how they differ from the interaction with MAC5A.

Thanks for pointing this out. We provided the following explanation in the text: "Interestingly, MAC5 formed speckles with CWC15 while CDC3 and MAC3A did not. A possible explanation is that MAC5, but not MAC3A and CDC5, contains unstructured region, which is known to promote speckles formation through liquid-liquid phase separation.". If the referee has additional comments, we are happy to follow.

2, In Figure 1d, the expression levels of miRNAs were determined with a certain qRT-PCR method, however, it does not seem to monitor the miRNA expression but rather seems to monitor precursors because the method description did not reflex the proper method used. To ensure that the miRNA expression levels presented in Figure 1d are indeed reflective of mature miRNAs, the authors should clearly state the specific miRNA qRT-PCR method used and provided. This information is crucial for a reliable and accurate assessment of miRNA expression in the study.

Appreciated the suggestion. We revised the description as following "In addition, stem-

loop qRT-PCR⁵² analysis showed that the reduced miRNA levels were fully recovered in the complementation lines (Fig. 1d), suggesting that CWC15 is required for miRNA accumulation.”.

3, In Figure 1e, it would be beneficial for the authors to include the median value of miRNA expression to facilitate the discussion of the overall reduction ratio and whether CWC15 functions as a general positive factor or a selectively functional factor.

Thanks for the suggestion. The value was added in the text.

4. In supplementary Figure 3, the expression levels of GFP-CWC15 should be presented to confirm that the complementation line.

Thanks for the suggestion. This line was also used to test the interaction of CWC15 with HYL1 and SE. Thus, we showed the western detection of GFP-CWC15 in Fig. 3d. We apologize that we did not describe it clearly. We added the description at the text when describing the Co-IP.

5, In Figure 2c-2f, the CWC15-dependent modulation of the miR167a promoter is monitored using GUS analysis and qRT-PCR. However, to confirm the specific interaction between CWC15 and the miR167a promoter, it is essential to include ChIP (Chromatin Immunoprecipitation) data.

Thanks for the suggestion. The ChIP data showing the interaction of CWC15 with MIR167a promoter shown in Fig. 2f.

6, In the processing assay, the size of the cleaved intermediates between MIR162b and pre-miR162b appears to be different. To confirm whether these are the same fragments, it would be beneficial to include RNA size markers in the analysis.

Thanks for pointing this out. Pri-miR162b is processed from loop-to-base. The intermediates may reflect the length of stem strand after removing loop. Thus the size of processing intermediate from MIR162b and pre-miR162b is different. The size of mature miR162 has been validated using radioactive marker from previous publication (PNAS 110: 17588-17593). We also did the experiments using Col with size marker (see below). If needed, this data will be included the manuscript.

7, The authors' claim that CWC15 is involved in SE degradation (Figure 5) may face challenges based on the observations in Figure S5. The transcript level of SE was

notably increased by over two-fold, and the Chase assay indicated that deficiency of CWC15 extended the half-life of SE by only 10 minutes. However, a reconstituted 20S proteasome degradation assay demonstrated a more substantial extension of SE half-life to approximately 35 to 40 minutes (Figure 5f). This suggests that the up-regulation of SE transcripts could be a contributing factor to the result observed in Figure 5c. To clarify this point, the authors should provide further explanation and delve into the relationship between the increased transcript levels and the extended SE half-life, addressing any potential mechanisms that might be responsible for the observed effects. We appreciate the wonderful suggestion. We fully agree the fact that increased SE transcript level may contribute to increase SE protein levels. We added this in the discussion section (second paragraph). However, CWC15 also greatly contributed to SE stability. In the Chase assay, translation is blocked by CHX (Cycloheximide), which should avoid the effect of transcription or translation rate. The Chase assay used protein extracts (low 20S proteasome concentration) from Col or amiR^{CWC15} (Fig, 5C), while reconstituted assay used purified 20S proteasome (high concentration). Thus, the concentration of 20S proteasome is different in these two assays, which makes it difficult to compare the decay rate of SE in two assays. We hope this could address the comments from the referee. We are happy to follow additional suggestions from the referee.

8, In Supplementary Figure S6d, there appears to be an error in the image-Q. The sample treated for 20 minutes shows a much lower SE (signal intensity) level compared to the sample treated for 30 minutes. However, the quantitation values for both samples are reported as the same, which seems inconsistent.

Really appreciate for pointing this out. We redid the quantification, see new Figure S6d.

9, In Figure 5, the authors stated that HYL1 is not altered in amiRCWC15 (Figure 5a). However, upon careful observation, it appears that the HYL1 protein level seems to exhibit a slight accumulation in Figure 5g (Total). Additionally, DCL1 also appears to show a slight increase in the artificial mutants in Figure 5a. These findings might raise questions and warrant further explanation from the authors. To address this, the authors should provide a more detailed explanation of these specific findings or present clearer and more conclusive results that align with their descriptions in the figure.

Thanks for the suggestion. We agree with that in Fig 5a, DCL1 may have slight increase, but not as significant as SE. We plan to investigate the mechanism in future studies. We examined the effect of CWC15 on HYL1 degradation, and did not observe significant effect (see below). In Fig. 5g, it may due to a loading difference. The goal of this experiment is to show effect of CWC15 on the distribution of SE in cytoplasmic and nucleus with HYL1 as control. Indeed, we did not observe the effect of CWC15 on HYL1 distribution. We are happy to follow additional advice on this.

10, CWC15 is a splicing-related gene. Does CWC15 influence the splicing of miRNA or other genes, like SEAP1(SERRATE-associated protein)?

Thanks for the suggestion. CWC15 indeed influence mRNA splicing, which as been reported by others (*Sci Rep-Uk* **10**, 13336). It also affects pri-miRNA splicing (Supplementary Fig. 4b).

11, In supplementary Figure 5, it would be beneficial to include the P-value in the legend. This addition will provide essential statistical information and enhance the clarity of the results presented in the figure.

Thanks for the suggestion, Added. See new legend of Supplementary Fig. 5.

12, Indeed, the localization of CWC15 in the nucleus is consistent with its role as a splicing factor. The BiFC assay also supports this nuclear localization, suggesting its involvement in nuclear processes. However, the detection of an association between CWC15 and the 20S proteasome in the nucleus raises questions regarding the regulation of SE degradation by CWC15 in the cytoplasm. Considering that impaired 20S proteasome activity leads to the accumulation of SE in the cytoplasm, it becomes less clear how CWC15, which is primarily localized in the nucleus, might be involved in regulating SE degradation in the cytoplasm. This observation raises the need for further investigations to better understand the interplay between CWC15, SE, and the 20S proteasome in different cellular compartments. Additional experiments, such as Co-IP assay in specific subcellular fractions, could help to shed light on the mechanisms involved in the regulation of SE degradation and how CWC15 might play a role in this process.

Thanks for the excellent point. Previous study showed that the 20S proteasome (nature plants, 6:970-982) and PRP4KA interact with SE in the nucleus (*Science Advance*, **8**: eabm8435), suggesting that SE is phosphorylated by PRP4KA in nucleus and subsequently degraded by 20S proteasome at the nucleus. In these studies, SE also accumulated in the cytoplasm in *pag-1*. It was proposed that the overaccumulation of SE is due to the reprogramming of protein trafficking genes in *pag-1*.

Agreeing with previous findings, our BiFC analysis showed that CWC15 mainly interacts with SE, PRP4K1A, and 20S proteasome in the nucleus, suggesting that

CWC15 majorly promote SE degradation in nucleus. In the revised manuscript, we performed cell fractionation assay, which shows that CWC15 localized at both cytoplasm and nucleus (Supplementary Fig. 6). Thus, as the referee suggested that CWC15 may also promote 20S proteasome activity on SE in cytoplasm. Alternatively, it may help 20S proteasome to affect protein trafficking genes, leading to the accumulation of SE in cytoplasm. We discussed this possibility in the revised manuscript.

However, the analysis the interplay among SE, 20S proteasome and CWC15 in different compartments is not a trivial task, which involves in cell biological analysis, mutation analysis, biochemical analysis (activity, co-IP and others) from cytoplasm or nucleus. Thus, we plan to perform the detailed study as an independent study, which we hope the referee will agree on this.

13, The recent study reporting the dual role of the HOS15/HDA9 complex in both promoter activity and pri-miRNA processing under ABA may add relevance to the introduction and could potentially support the role of CWC15 and its additional role in SE proteostasis.

Thanks for the suggestion. This was added in the introduction section.

Reviewer #2 (Remarks to the Author):

In this manuscript, Yu and colleagues present novel roles of CWC15 in microRNA biogenesis in plant. Previously identified as a spliceosome associated factor, CWC15 is now demonstrated to play a pivotal role in promoting miRNA generation through multiple gene regulatory mechanisms.

The authors revealed that CWC15 binds to the promoters of MIR genes and helps to recruit polymerase II. Furthermore, CWC15 interacts with the DCL1 complex and enhances the RNA binding affinity of HYL, a key component of the DCL1 complex. This enhancement results in more effective microRNA processing. The study also uncovers a role of CWC15 in maintaining D-bodies, specialized structures involved in microRNA processing. CWC15 destabilizes the SE protein in the cytoplasm through the 20S proteasome and regulates the appropriate amount of SE proteins within the D-bodies. Interaction of CWC15 with PRP4KA leads to the phosphorylation of the SE protein, subsequently inducing degradation of SE protein.

This study effectively elucidates the biological significance of CWC15 in plant microRNA biogenesis very well. However, the following concerns need to be adequately addressed in order to publish this study in Nature Communications.

-Major Concerns

1. The question arises whether CWC15 is also involved in microRNA biogenesis in other eukaryotes. Addressing or discussion whether the roles of CWC15 in microRNA biogenesis are specific to plants or conserved across eukaryotes would be highly valuable.

Appreciate the suggestion. We added a discussion in the last paragraph of discussion

section as following “We suspect that like in Arabidopsis, CWC15 might also influence the production of miRNAs in other organisms by modulating orthologs of SE, such as the mammalian ARS2. Supporting this notion, the null mutations of CWC15 causes embryo lethality of mammals, which agrees with essential of miRNA in controlling development.”.

2. While the expectation is for CWC15 to reside in the nucleus, it is intriguing that CWC15 regulates SE proteins in the cytoplasm through PRP4KA and the 20S proteasome. Therefore, it would be essential to demonstrate the subcellular localization of CWC15 in both the nucleus and cytoplasm to gain further insights into its versatile functions.

Thanks for the suggestion. We examined the localization of CWC5 in cytoplasm and nucleus and the result showed that CWC5 localized both compartments (Supplementary Fig. 7).

3. The authors generated three lines of amiRCWC15 (amiRCWC15 -1, -2 and -3). However, for most experiments, amiRCWC15 -1 was primarily utilized. Only for northern blot analysis, amiRCWC1-2 was additionally employed. It would be beneficial to observe the consistency of experimental results in the other amiRCWC15 lines, similar to the trends shown in the experimental results presented for amiRCWC15 -1.

Thanks for the suggestion. We agree with the referee that additional lines will be beneficial. We used both line 1 and line 2 for many experiments, which include the expression levels of, miRNAs (Fig 1a), pri-miRNAs (Fig. 2a) and genes involved in miRNA biogenesis showed similar trend (protein levels in Fig 5a, transcript levels in Fig S4). In addition, the expression of amiR^{CWC15} resistant CWC15 complemented the phenotype of line 1 and line 2. We apologize that we did not describe these clearly. We think that these results should be sufficient to support our conclusion, which we hope the referee will agree. However, we are happy to follow additional advices.

4. It is unclear whether a single microRNA can be regulated by CWC15 at all stages of biogenesis from transcription to posttranscriptional cleavage. One question that arises is whether CWC15 selectively operates at specific stages of biogenesis depending on the features of the miRNA genes. While the study investigates a list of microRNAs to verify the roles of CWC15 in microRNA biogenesis, none of the microRNA genes were utilized in all experiments. For instance, miR-167 was detected by northern blotting and the expression change of its target gene was validated. Expression levels of mature miR-167a and pri-miR-167a were measured by qRT-PCR and its association with HYL1 was shown (Fig. 4h). However, the association of PolIII with miR-167 promoter was not examined in ChIP experiments. Another example is miR-159a. Expression of miR-159a was measured by northern blotting and qRT-PCR. In PolIII-ChIP experiments, miR-159a promoter was detected (Fig. 2d), but it was not examined in a rescue experiment (Fig. 2f). In addition, in HYL-RIP experiments, miR-159a was not tested either (Fig. 4h). To provide a clearer understanding of whether a single microRNA can

be regulated at all layers of biogenesis by CWC15, it would be beneficial to investigate multiple microRNAs and examine their regulation through all stages of biogenesis. By addressing this aspect, it can be better determined whether CWC15 indeed regulates microRNA expression at all stages of biogenesis.

Really appreciate the excellent points. We now showed the levels of pri-miR156a, pri-miR159a, pri-miR167a and pri-miR172b are reduced. Consistent with this observation, ChIP assay showed that Pol II occupancy at their promoters was reduced and CWC15 binds their promoters. Moreover, RIP assay showed that HYL1 interaction with these pri-miRNAs are reduced. These results suggest that CWC15 may affect multiple steps of miRNA biogenesis.

5. In Fig. 1e and Data set1, it is described that miRNA expression levels are decreased in *amiRCWC15 -1*. However, it is worth noting that out of a total 354 miRNAs, 260 miRNAs are downregulated and 93 miRNAs are upregulated in *amiRCWC15 -1*. This raised the question of whether there are specific features in the sequence or structure of the upregulated miRNAs that may explain the different patterns. To generalize the reduction of miRNAs in *amiRCWC15 -1*, the data is from biological replicates should be presented with the result of statistical analysis. Furthermore, a detailed discussion on miRNAs produced from both arms of a common precursor but showing different accumulation patterns in *amiRCWC15 -1* is necessary. For example, miR-171c-3p decreased while miR-171-5p increased in *amiRCWC15 -1*. Similar to this, miR-167a-3p and miR-167c-3p are upregulated while their corresponding -5p miRNAs are downregulated in *amiRCWC15 -1*.

Thanks for the excellent suggestion. We added the P-value in the Dataset S1. We examined the pri-miRNA structure/sequence of differential regulated miRNAs. However, we did not find any specific features. We suspect that the upregulation of some miRNAs in may due to the function in splicing of CWC15, which may affect the expression levels of transcription factors that control the expression of specific miRNAs. Supporting this notion, the accumulation of some miRNAs also is increased in other *mac* mutants. We added the description in the discussion section.

We also discussed the differential effect of CWC15 on two arms from a common precursor as following “We noticed that CWC5 differentially affects the accumulation of miRNAs produced from two arms of a common pri-miRNA. For instance, the abundance of miR-171c-3p, miR-167a-5p and miR-167c-5p was decreased in *amiR^{CWC15}*, while the amount of miR-171-5p miR-167a-3p and miR-167c-3p were increased in *amiR^{CWC15}*. Studies have revealed that the differential accumulation of two arms from the same precursor is controlled at processing or post-processing levels. Given the association of CWC15 with the DCL1 complex, we suspect that CWC15 may play a role in modulating the arm-specific processing of some pri-miRNAs.”.

- Minor Concerns

6. For general readers, it would be beneficial to include a brief description of CDC5 at

the beginning of Result section. CDC5 is a positive regulator of pri-miRNA transcription and processing (Shuxin Zhang et al, 2007, PNAS). annotation for CWC15 throughout the manuscript.

Thanks for the suggestion. We added the description.

7. In text (line 137), CWC15 is annotated as AT3G13200, whereas it is annotated as EMB2769 in Sup Fig. 1.

Thanks for the suggestion. We added EMB2769 in the description of description.

8. In text (line 207), AGO1 is mentioned, but in Fig. 3 DDL is presented.

Thanks for the suggestion. We revised our description accordingly.

9. In Fig. 4c, d, it would be helpful to include size marker on a gel and provide an explanation for the band between the substrate and mature miRNA. It is noteworthy that the intermediate-sized bands are produced even when protein extract from amiRCWC15 were used, despite mature miRNAs not being produced.

Thanks for pointing this out. miR162b likely is processed from loop-to-base. The intermediates may reflect the length of each strand of stem after removing loop. Thus the size of processing intermediate from MIR162b and pre-miR162b is different. The size of mature miR162 has been validated using radioactive marker from previous publication (PNAS 110: 17588-17593). We also repeated the experiments using Col with size marker (see below). If needed, this data will be included the manuscript.

10. In Fig. 4g, the efficiency of HYL-IP seems to be relatively low. To improve the accuracy of the results, it would be advisable to use IgG as a control for HYL IP instead of employing no antibody.

Thanks for the suggestion. The low IP signal may be due to the amount difference of sample loading between IP (equal to 1/160 input) and input (1/40 of input). In addition, this commercial HYL1 antibody has been used by several labs for RIP. Thus, given the fact that the HYL1-RNA interaction is significantly reduced in *cwc15* relative Col, we wish to keep the current result. We fully agree with the referee that IgG is a better control, and we will follow the suggestion in the future experiments.

REVIEWER COMMENTS

Reviewer #1 (Remarks to the Author):

The authors have addressed and provided explanations for the raised discussions. Two additional points have been suggested for inclusion in the article. First, the processing assay presented in the "response to Referee letter" should be incorporated. Second, a reference, "Jung et al., PNAS 2022, doi: 10.1073/pnas.2116757119," should be added to the introduction, specifically in lines 95-100, where the regulations of HYL1 stability were mentioned.

Reviewer #2 (Remarks to the Author):

The authors have adequately addressed most of the questions. However, a few additional revisions are suggested:

Regarding the additional lines of amiRCWC15, it is recommended to display the real-time PCR results for the target mRNA levels in amiRCWC15-2, similar to Figure 1f.

Additionally, it is strongly recommended to demonstrate the restoration of target gene expression levels in GFP-CWC15/amiRcwc15, as shown in Figure 1.

In Figures 4c and 4d, it is advisable to mark the intermediate-size bands and provide a description of these bands in the figure legends

Point-by-Point Response to Reviewer's comments

We really appreciate that most comments have been addressed. We are very grateful for the additional comments. The changes in the text are highlighted in Blue.

Reviewer #1 (Remarks to the Author):

The authors have addressed and provided explanations for the raised discussions. Two additional points have been suggested for inclusion in the article.

We are happy that most of the comments have been addressed.

First, the processing assay presented in the "response to Referee letter" should be incorporated. Thanks for the advice. The data was added as supplementary Fig. 5 and the size difference of the intermediates was explained in the legends.

Second, a reference, "Jung et al., PNAS 2022, doi: 10.1073/pnas.2116757119," should be added to the introduction, specifically in lines 95-100, where the regulations of HYL1 stability were mentioned.

Thanks for the advice. The reference was added.

Reviewer #2 (Remarks to the Author):

The authors have adequately addressed most of the questions. However, a few additional revisions are suggested:

We are happy that most of the comments have been addressed.

Regarding the additional lines of amiRCWC15, it is recommended to display the real-time PCR results for the target mRNA levels in amiRCWC15-2, similar to Figure 1f.

Thanks for the advice. The result was added in figure 1.

Additionally, it is strongly recommended to demonstrate the restoration of target gene expression levels in GFP-CWC15/amiRcwc15, as shown in Figure 1.

Thanks for the advice. The result was added in figure 1.

In Figures 4c and 4d, it is advisable to mark the intermediate-size bands and provide a description of these bands in the figure legends

Thanks for the advice. A size mark was added in the figures. The size difference of the intermediates in two experiments were added and explained in the figure legend of supplementary figure 5.